# An unexpected role of neutrophils in clearing apoptotic hepatocytes in vivo

Luyang Cao[1,2†], Lixiang Ma[3†], Juan Zhao[1†], Xiangyu Wang[4†], Xinzou Fang[1], Wei Li[2], Yawen Qi[2], Yingkui Tang[1], Jieya Liu[1], Shengxian Peng[1], Li Yang[1], Liangxue Zhou[1], Li Li[4], Xiaobo Hu[5], Yuan Ji[6], Yingyong Hou[6], Yi Zhao[7], Xianming Zhang[8], You-yang Zhao[8], Yong Zhao[9], Yuquan Wei[1], Asrar B Malik[10], Hexige Saiyin[11]*, Jingsong Xu[1*‡]

[1]Department of Neurosurgery, State Key Laboratory of Biotherapy and Cancer Center, West China Hospital, Sichuan University, Chengdu, China; [2]Guangzhou Regenerative Medicine and Health Guangdong Laboratory (GRMH-GDL), Guangzhou, China; [3]Department of Anatomy, Histology & Embryology, Shanghai Medical College, Shanghai, China; [4]Department of Anatomy, Histology & Embryology, Shanghai Medical College, Shanghai, China; [5]Clinical Laboratory, Longhua Hospital, Shanghai University of Traditional Medicine, Shanghai, China; [6]Department of Pathology, Zhongshan Hospital Fudan University, Shanghai, China; [7]Department of Rheumatology and Immunology, West China Hospital, Sichuan University, Chengdu, China; [8]Program for Lung and Vascular Biology, Stanley Manne Children's Research Institute, Ann & Robert H. Lurie Children's Hospital of Chicago, and Department of Pediatrics, Division of Critical Care, Northwestern University Feinberg School of Medicine, Chicago, United States; [9]State Key Laboratory of Membrane Biology, Institute of Zoology, Chinese Academy of Sciences, Beijing, China; [10]Department of Pharmacology, University of Illinois, College of Medicine, Chicago, United States; [11]State Key Laboratory of Genetic Engineering, School of Life Sciences, Fudan University, Shanghai, China

*For correspondence: 18621117074@163.com (HS); jingsong.xu@hotmail.com (JX)

†These authors contributed equally to this work

Present address: ‡Department of Pharmacology, Center for Lung and Vascular Biology, University of Illinois, Chicago, United States

Competing interest: The authors declare that no competing interests exist.

**Abstract** Billions of apoptotic cells are removed daily in a human adult by professional phagocytes (e.g. macrophages) and neighboring nonprofessional phagocytes (e.g. stromal cells). Despite being a type of professional phagocyte, neutrophils are thought to be excluded from apoptotic sites to avoid tissue inflammation. Here, we report a fundamental and unexpected role of neutrophils as the predominant phagocyte responsible for the clearance of apoptotic hepatic cells in the steady state. In contrast to the engulfment of dead cells by macrophages, neutrophils burrowed directly into apoptotic hepatocytes, a process we term *perforocytosis*, and ingested the effete cells from the inside. The depletion of neutrophils caused defective removal of apoptotic bodies, induced tissue injury in the mouse liver, and led to the generation of autoantibodies. Human autoimmune liver disease showed similar defects in the neutrophil-mediated clearance of apoptotic hepatic cells. Hence, neutrophils possess a specialized immunologically silent mechanism for the clearance of apoptotic hepatocytes through perforocytosis, and defects in this key housekeeping function of neutrophils contribute to the genesis of autoimmune liver disease.

## eLife assessment

This paper reports the **fundamental** discovery of a new function of neutrophil in specifically clearing apoptotic hepatocytes by penetrating the cells rather than engulfing them without causing

inflammation as a part of tissue homeostasis. This **solid** study transforms the way we think about role of neutrophil in pathogenesis of autoimmune liver disease.

## Introduction

Apoptosis is a process of programmed cell death that clears aged or damaged cells to maintain internal tissue homeostasis (*Hochreiter-Hufford and Ravichandran, 2013*). An estimated hundred billion cells undergo apoptosis daily in a human adult (*Fond and Ravichandran, 2016*). These apoptotic cells must be disposed of promptly and efficiently without littering cytoplasm or causing inflammation (*Franz et al., 2006*). Defects in the clearance of apoptotic bodies are often linked to various inflammatory and autoimmune diseases (*Nagata et al., 2010*; *Poon et al., 2014*). It is widely believed that apoptotic bodies are cleared by professional phagocytes, such as macrophages and immature dendritic cells, or by local nonprofessional phagocytes, such as epithelial cells, endothelial cells, and fibroblasts (*Lauber et al., 2004*). Although the morphology of apoptotic cells is easily distinguished in histological sections (*Kerr et al., 1972*), these cells are infrequently observed in normal human samples due to efficient removal by phagocytes (*Lauber et al., 2004*; *Poon et al., 2014*). Hence, the phagocytes responsible for the removal of apoptotic cells in the homeostatic state remain uncertain and it is unknown whether they are tissue specific.

Neutrophils, a type of terminally differentiated and short-living phagocytic cell, represent 50–70% of the total white blood cell population in humans (*Amulic et al., 2012*; *Kolaczkowska and Kubes, 2013*). However, neutrophils are not considered key players in apoptotic cell clearance (*Nagata et al., 2010*; *Poon et al., 2014*). They instead function as inflammatory cells responsible for killing bacteria and fighting infection. Blood neutrophils often swarm to the site of infection, where they release toxic mediators (e.g. reactive oxygen species and proteases) that not only kill microorganisms but also can damage tissues (*Amulic et al., 2012*; *Kolaczkowska and Kubes, 2013*). To prevent inflammation, apoptotic cells have been shown to release 'keep-out' signals, such as lactoferrin, that prevent the recruitment of neutrophils to the apoptotic site (*Bournazou et al., 2009*). Nevertheless, neutrophils are related to multiple autoimmune diseases (*Amulic et al., 2012*; *Kolaczkowska and Kubes, 2013*). For example, mild neutropenia was observed to precede and accompany the onset of type 1 diabetes, a typical autoimmune disease (*Harsunen et al., 2013*; *Valle et al., 2013*). The role of neutrophils in the clearance of apoptotic bodies and contribute to autoimmunity remain to be solved (*Jorch and Kubes, 2017*).

Here, we report an unexpected role of neutrophils in clearing apoptotic hepatocytes under physiological conditions. We found that neutrophils, not macrophages or other cells, are responsible for apoptotic hepatocyte clearance in an immunologically silent manner. We noted that neutrophils penetrate apoptotic hepatocytes and clear them from the inside, thereby avoiding spillage of cytoplasmic content; this mechanism differs sharply from the classical engulfment of dead cells by macrophages. Defects in neutrophil-mediated apoptotic clearance have been observed in human autoimmune liver (AIL) disease. Hence, in addition to their well-known role in combating infection, neutrophils can function as housekeepers for apoptotic clearance and thus maintain tissue homeostasis.

## Results

### Neutrophils burrow into apoptotic hepatocytes in human livers

We observed a large number of neutrophils in hepatocytes in human noncancerous liver tissue obtained from patients with hepatocellular carcinoma (*Figure 1A*) or from patients with hepatic hemangioma (*Figure 1B*). We discovered that the hepatocytes occupied by neutrophils were apoptotic, as evidenced by the condensed chromatin signature (*Figure 1A*, panels i–vi, black arrowheads indicate condensed chromatin, white arrowheads point to neutrophils with a characteristic multilobed nucleus). Importantly, apoptotic hepatocytes are rarely observed in the human liver, possibly due to the rapid removal of apoptotic cells by phagocytes. In a total of 281 apoptotic hepatocytes from 32 livers, we observed that each apoptotic hepatocyte was engorged by up to 22 neutrophils (*Figure 1C*, *Table 1*). We also confirmed apoptotic hepatocytes by TUNEL staining or Caspase-3 immunostaining (*Figure 1D*). Neutrophils burrowed inside apoptotic hepatocytes were either stained with an antibody against neutrophil elastase (NE), myeloperoxidase, or recognized by their multilobed nucleus

**eLife digest** Every day, the immune cells clears the remains of billions of old and damaged cells that have undergone a controlled form of death. Removing them quickly helps to prevent inflammation or the development of autoimmune diseases.

While immune cells called neutrophils are generally tasked with removing invading bacteria, macrophages are thought to be responsible for clearing dead cells. However, in healthy tissue, the process occurs so efficiently that it can be difficult to confirm which cells are responsible.

To take a closer look, Cao et al. focused on the liver by staining human samples to identify both immune and dead cells. Unexpectedly, there were large numbers of neutrophils visible inside dead liver cells. Further experiments in mice revealed that after entering the dead cells, neutrophils engulfed the contents and digested the dead cell from the inside out. This was a surprising finding because not only are neutrophils not usually associated with dead cells, but immune cells usually engulf cells and bacteria from the outside rather than burrowing inside them.

The importance of this neutrophil behaviour was shown when Cao et al. studied samples from patients with an autoimmune disease where immune cells attack the liver. In this case, very few dead liver cells contained neutrophils, and the neutrophils themselves did not seem capable of removing the dead cells, leading to inflammation. This suggests that defective neutrophil function could be a key contributor to this autoimmune disease.

The findings identify a new role for neutrophils in maintaining healthy functioning of the liver and reveal a new target in the treatment of autoimmune diseases. In the future, Cao et al. plan to explore whether compounds that enhance clearance of dead cells by neutrophils can be used to treat autoimmune liver disease in mouse models of the disease.

signature (*Figure 1D*, *Figure 1—figure supplement 1A*). The distances from burrowed neutrophils to the apoptotic hepatocyte border were analyzed by IMARIS software and recorded in *Table 2*. Other immune cells were rarely associated with apoptotic hepatocytes. Kupffer cells are liver-resident phagocytes that are thought to clear apoptotic bodies (*Canbay et al., 2003*; *Eipel et al., 2007*). Upon staining with an antibody against CD68, a marker of Kupffer cells, we observed few Kupffer cells invading or engulfing apoptotic hepatocytes (*Figure 1C,E*, 7A and *Table 3*). Furthermore, we detected few CD11b[+] or CD45RA[+] cells associated with the apoptotic sites (*Figure 1—figure supplement 1B*, *Tables 4 and 5*).

Based on the above observations that neutrophils are the predominant phagocyte associated with apoptotic hepatocytes, we hypothesized that the neutrophil-mediated clearance of apoptotic cells consists of the following three sequential steps. In the initial invading or burrowing stage, activated neutrophils identified and targeted the apoptotic hepatocytes (cells with condensed chromatin, indicated by black arrowheads) and attached to their cell membrane (*Figure 1A*, panel i, outlined by black rectangle). Then, the neutrophils invaded apoptotic hepatocytes (*Figure 1A*, panels ii and iii). We observed an average of seven neutrophils entering each apoptotic hepatocyte, and we termed this process perforocytosis (from Latin *perfero* to bore) (*Figure 1A*, white arrowheads point to neutrophils). The second step consisted of phagocytosis and detachment. The neutrophils within hepatocytes appeared to clear apoptotic bodies from the inside without destroying the cellular membrane or extruding the cytoplasm (two burrowed neutrophils phagocytosed apoptotic debris as shown in *Figure 1—figure supplement 1C*). Following digestion by neutrophils, the apoptotic hepatocytes decreased in size, and detached from nearby hepatocytes (*Figure 1A*, panels iv–vi). The third step involved the complete digestion of apoptotic hepatocytes (*Figure 1A*, panels vii–ix). After the clearance of apoptotic hepatocytes, neutrophils seemed to migrate away from the cleared space, possibly to make room for new hepatocytes generated by rapid division. This 'eating inside' phagocytosis process differed sharply from the well-known engulfment of apoptotic cells or fragmented apoptotic bodies by other phagocytic cells, such as macrophages (*Poon et al., 2014*).

## Visualization of neutrophil perforocytosis in mouse livers

Neutrophils within apoptotic hepatocytes were confirmed in mouse livers by intravital microscopy (*Figure 2A and B*) and electron microscopy (*Figure 2C*). Similar to the observations in human

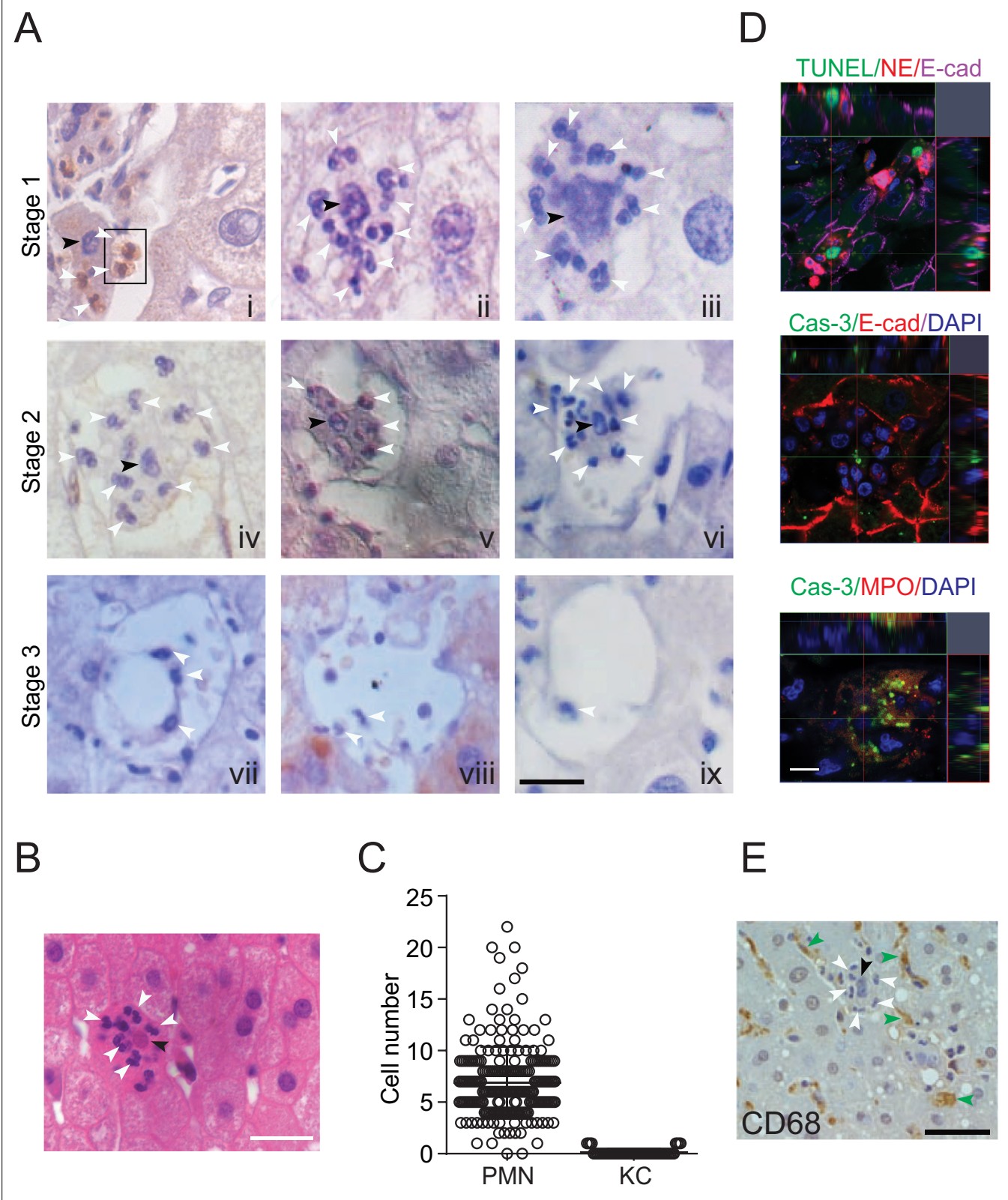

**Figure 1.** Neutrophil-mediated clearance of apoptotic cells in livers. (**A**) Hematoxylin staining of human noncancerous liver tissues from patients with hepatocellular carcinoma. Apoptotic hepatocytes with apparent condensed chromatin are being scavenged by several neutrophils (black arrowhead indicates condensed chromatin and white arrowheads point to neutrophils with a characteristic multilobed nucleus). There are three stages of neutrophil scavenging apoptotic hepatocytes in human samples. Stage 1 (i–iii), initial invading stage: two neutrophils start to invade an apoptotic hepatocyte with

*Figure 1 continued on next page*

*Figure 1 continued*

apparent condensed chromatin (outlined by the black rectangle); three other neutrophils are already inside the cytoplasm (**i**); at this stage, the apoptotic hepatocytes are still attached to neighboring hepatocytes and occupied by 2–16 neutrophils (**ii–iii**, 37 out of 241 apoptotic hepatocytes observed are at stage 1). Stage 2, phagocytosis and shrinking stage: The apoptotic hepatocytes are being phagocytosed by invading neutrophils and are partially detached from neighboring hepatocytes (**iv–vi**, 130 out of 241 apoptotic cells are at stage 2). Stage 3, complete digestion stage: apoptotic hepatocytes are completely detached and largely cleared, and only neutrophils remain in the cleared region (**vii–ix**, 74 out of 241 apoptotic cells are at stage 3). Scale bar, 20 µm. (**B**) Hematoxylin staining of human liver tissues from patients with hemangioma (a total of 40 apoptotic hepatocytes were observed). Scale bar, 20 µm. (**C**) Cell counts of neutrophils (polymorphonuclear neutrophils [PMNs]) and Kupffer cells (KCs) inside or associated with apoptotic hepatocytes, also see *Tables 1 and 3*. (**D**) Fluorescent images of human liver tissues with neutrophils scavenging apoptotic hepatocytes. Apoptotic hepatocytes are confirmed with TUNEL staining or Caspase-3 immunostaining. Neutrophils are labeled with immunostaining of neutrophil elastase (NE), myeloperoxidase (MPO), or DAPI staining (with a segmented nucleus signature). Scale bar, 10 µm. (**E**) Images of liver tissue stained with an antibody against CD68, a marker of KCs (indicated by green arrowheads). KCs do not invade or phagocytose apoptotic hepatocytes (black arrowheads). Data are representative of (**A, B, D, E**) or from (**C**) a total of 281 apoptotic hepatocytes observed, mean and sem in (**C**).

The online version of this article includes the following figure supplement(s) for figure 1:

**Figure supplement 1.** Little leakage during perferocytosis.

samples, apoptotic hepatocytes in wild-type (WT) mouse livers were occupied by neutrophils but not associated with macrophages (*Figure 2A*, neutrophils were labeled with an i.v. injection of anti-Ly6G antibody, macrophages were labeled with anti-F4/80 antibody and apoptotic cells were labeled with Annexin V). A total of 24 apoptotic cells were observed in eight WT livers with an average of two burrowed neutrophils in each mouse apoptotic hepatocyte. The burrowed neutrophils were projected and analyzed by IMARIS software and results were compared side by side with livers from MRP8cre/DTR mice (*Figure 2A*, with more details in the neutrophil depletion section below). The distances of burrowed neutrophils to the border of apoptotic hepatocytes were recorded in *Table 6*.

By using the Cellvizio System (Confocal Miniprobes), we managed to visualize the entire process of neutrophil perforocytosis in mouse livers (*Figure 2B* and *Videos 1 and 2*, apoptotic hepatocytes were labeled with Annexin V and neutrophils were labeled with anti-Ly6G antibody). Consistent with observations in human samples, Annexin V positive apoptotic hepatocytes were burrowed and cleared by Ly6G-labeled neutrophils (*Figure 2B*). This eating inside phagocytic process is fast and rigorous in which neutrophils were able to completely digest apoptotic hepatocytes around 4–7 min (*Figure 2B* and *Videos 1, 2*, a total of 13 apoptotic hepatocytes in 12 WT mouse livers were observed). At the end of the apoptotic clearance process, neutrophils simply left the apoptotic sites and were not labeled by Annexin V, indicating neutrophils were not apoptotic.

**Table 1.** The cell numbers of burrowed neutrophils (polymorphonuclear neutrophils [PMNs]) in apoptotic hepatocytes from human liver tissues.

2D images were taken from regular tissue sections (5 µm thick), and 3D images were taken from 45 µm thick tissue sections and reconstructed by confocal microscope.

| | Perferocytosis stage | I | II | III |
|---|---|---|---|---|
| | Apoptotic hepatocytes (total 241) | 37 | 130 | 74 |
| | Burrowing PMNs (total 1716) | 312 | 931 | 473 |
| 2D images | PMNs per apoptotic hepatocyte (mean±sem) | 8.4±4.2 | 7.2±3.4 | 6.4±3.4 |
| | Apoptotic hepatocytes (total 60) | 28 | 18 | 14 |
| | Burrowing PMNs (total 820) | 406 | 348 | 66 |
| 3D images | PMNs per apoptotic hepatocyte (mean±sem) | 14.5±10.0 | 19.3±15.5 | 4.7±2.4 |

**Table 2.** Analysis of neutrophils burrowing inside apoptotic hepatocytes from human liver samples. Reconstructed 3D images of neutrophils inside apoptotic hepatocytes (as shown in *Figure 1D*) are remodeled and analyzed with IMARIS software: anti-E-Cadherin blue staining is set as the source channel to detect hepatic cell boundary, and anti-NE red staining is set as the source channel to detect neutrophils inside blue cells. Only neutrophils inside hepatic cells can be detected. The cell positions of both neutrophils and hepatocytes, and the distances from neutrophils to the hepatocyte border are recorded below (a total of eight apoptotic hepatocytes are analyzed).

| Apoptotic hepatocytes | Hepatocyte position X,Y,Z (µm) | PMNs inside hepatocytes | PMN position X,Y,Z (µm) | PMN distance to hepatocyte border (µm) |
|---|---|---|---|---|
| | | #1 | 13.3, 3.7, 1.9 | 0 |
| | | #2 | 10.1, 4.1, 3.9 | 1.9 |
| | | #3 | 18.6, 9.8, 3.1 | 1.1 |
| | | #4 | 8.5, 12.2, 3.5 | 1.5 |
| | | #5 | 3.2, 22.6, 7.1 | 1.2 |
| #1 | 10.8, 10.6, 4.6 | #6 | 8.1, 22.7, 7.7 | 3.8 |
| | | #1 | 28.1, 21.0, 0.8 | 0 |
| | | #2 | 31.2, 18.9, 3.5 | 1.5 |
| #2 | 31.4, 14.7, 7.5 | #3 | 30.4, 5.1, 5.1 | 3.1 |
| #3 | 38.9, 15.1, 3.0 | #1 | 38.9, 15.2, 3.5 | 1.5 |
| #4 | 28.9, 24.1, 7.9 | #1 | 34.1, 19.4, 11.3 | 10.1 |
| | | #1 | 12.6, 33.2, 7.2 | 5.9 |
| | | #2 | 15.6, 35.5, 5.6 | 4.4 |
| | | #3 | 18.9, 37.9, 5.1 | 3.9 |
| #5 | 16.7, 35.4, 14.2 | #4 | 13.1, 36.5, 9.9 | 8.6 |
| | | #1 | 27.5, 21.9, 2.4 | 1.1 |
| #6 | 25.5, 22.8, 3.0 | #2 | 25.1, 23.2, 3.4 | 2.2 |
| | | #1 | 16.0, 43.0, 8.3 | 3.5 |
| #7 | 15.3, 44.1, 9.4 | #2 | 13.8, 45.4, 11.0 | 1.2 |
| #8 | 24.0, 35.4, 8.8 | #1 | 24.0, 35.7, 9.4 | 8.1 |

## The selectivity of neutrophil perforocytosis of effete hepatocytes

To study neutrophil-mediated phagocytosis in live cells in vitro, we induced apoptosis of isolated human primary hepatocytes or mouse liver NCTC cells with puromycin (*Figure 3—figure supplement 1A*) and then applied human neutrophils or neutrophil-like, differentiated HL60 cells (*Figure 3*).

**Table 3.** CD68[+] cells' distribution in human liver tissues with or without apoptotic hepatocytes (perforocytosed by polymorphonuclear neutrophils [PMNs]).
Five µm thick tissue sections were used to stain and count CD68[+] cells.

| | Total area (µm²) | CD68[+] cells | Apoptotic hepatocytes | CD68[+] cells/10⁴ µm² (mean±sem) |
|---|---|---|---|---|
| Liver tissue with PMN perforocytosis | 579,175 | 280 | 40 | 5.8±2.2[*] |
| Liver tissue without PMN perforocytosis | 403,025 | 267 | 0 | 6.7±2.8[*] |

[*]p=0.0962 (Student's *t*-test), CD68[+] cell numbers are not different in liver tissues with or without PMN perforocytosis.

**Table 4.** The distribution of CD11b+ cells in human liver tissues with or without apoptotic hepatocytes (perferocytosed by polymorphonuclear neutrophils [PMNs]).
45 μm thick tissue sections were used to stain and count CD11b+ cells.

| | Total area (μm²) | CD11b+ cells | Apoptotic hepatocytes | CD11b+ cells/10⁴ μm² (mean±sem) |
|---|---|---|---|---|
| Liver tissue with PMN perferocytosis | 138,905 | 12 | 17 | 1.1±1.4* |
| Liver tissue without PMN perferocytosis | 228,508 | 72 | 0 | 3.1±3.0* |

*p=0.0013 (Student's t-test), CD11b+ cell numbers are lower in liver tissues with PMN perferocytosis compared with that without PMN perferocytosis.

Isolated human primary hepatocytes formed a hepatic plate-like structure around 7 days in the culture dishes and were further confirmed with anti-albumin antibody (*Figure 3A*). Human blood neutrophils (labeled with a red membrane dye, PKH-26) were able to penetrate and burrow inside the human apoptotic hepatocytes induced by puromycin (*Figure 3B*, white arrowheads point to burrowing neutrophils). These apoptotic hepatocytes had markedly decrease in size after neutrophil burrowing as compared with cells without neutrophil burrowing (*Figure 3C and D*, *Video 3* shows a human neutrophil first burrowed and then started to phagocytose an apoptotic hepatocyte from inside in real time). Similar results were observed with NCTC cells (labeled with a red membrane dye, PKH-26) and HL60 cells (labeled with a green membrane dye, PKH-67). Green-labeled HL60 cells showed little response to normal NCTC cells (*Figure 3E*, top row). In the presence of apoptotic NCTC cells, however, green-labeled HL60 cells polarized and penetrated dead cells (*Figure 3E*, white arrowheads point to burrowing neutrophils, bottom row). Next, we quantified phagocytosis by flow cytometry using a pH sensitive dye, PHrodo (with weak fluorescent at neutral pH but high fluorescent with a drop in pH to measure engulfment, see Methods). The phagocytosis of apoptotic hepatocytes by neutrophils was markedly greater than that of normal nonapoptotic hepatocytes (*Figure 3F and G*). There was little cytoplasm or DNA leakage during neutrophils phagocytosing apoptotic NCTC cells as assessed by the extracellular levels of DNA, SOD, and ROS (*Figure 3—figure supplement 1D–F*). To address whether neutrophils also phagocytose other apoptotic cells, we determined the ability of neutrophils to phagocytose apoptotic endothelial HUVECs or epithelial HEK293 cells. We did not observe phagocytosis of these cells compared with their nonapoptotic controls (*Figure 3G*), indicating the selectivity of neutrophils in phagocytosing apoptotic hepatocytes. Upon comparing the ability of HL60 cells and macrophage U937 cells to phagocytose apoptotic hepatocytes, we found that U937 cells exhibited much lower phagocytosis of apoptotic NCTC cells than did HL60 cells (*Figure 3—figure supplement 1C*). Similar results were observed with human and mouse macrophages (*Figure 3—figure supplement 1B, C*), consistent with the central role of neutrophils in mediating apoptotic hepatocyte clearance.

**Table 5.** The distribution of CD45RA+ cells in human liver tissues with or without apoptotic hepatocytes (perferocytosed by polymorphonuclear neutrophils [PMNs]).
45 μm thick tissue sections were used to stain and count CD45RA+ cells.

| | Total area (μm²) | CD45RA+ cells | Apoptotic hepatocytes | CD45RA+ cells/10⁴ μm² (mean±sem) |
|---|---|---|---|---|
| Liver tissue with PMN perferocytosis | 57,367 | 7 | 10 | 1.4±1.2* |
| Liver tissue without PMN perferocytosis | 280,880 | 54 | 0 | 1.9±1.9* |

*p=0.309 (Student's t-test), CD45RA+ cell numbers are not different in liver tissues with or without PMN perferocytosis.

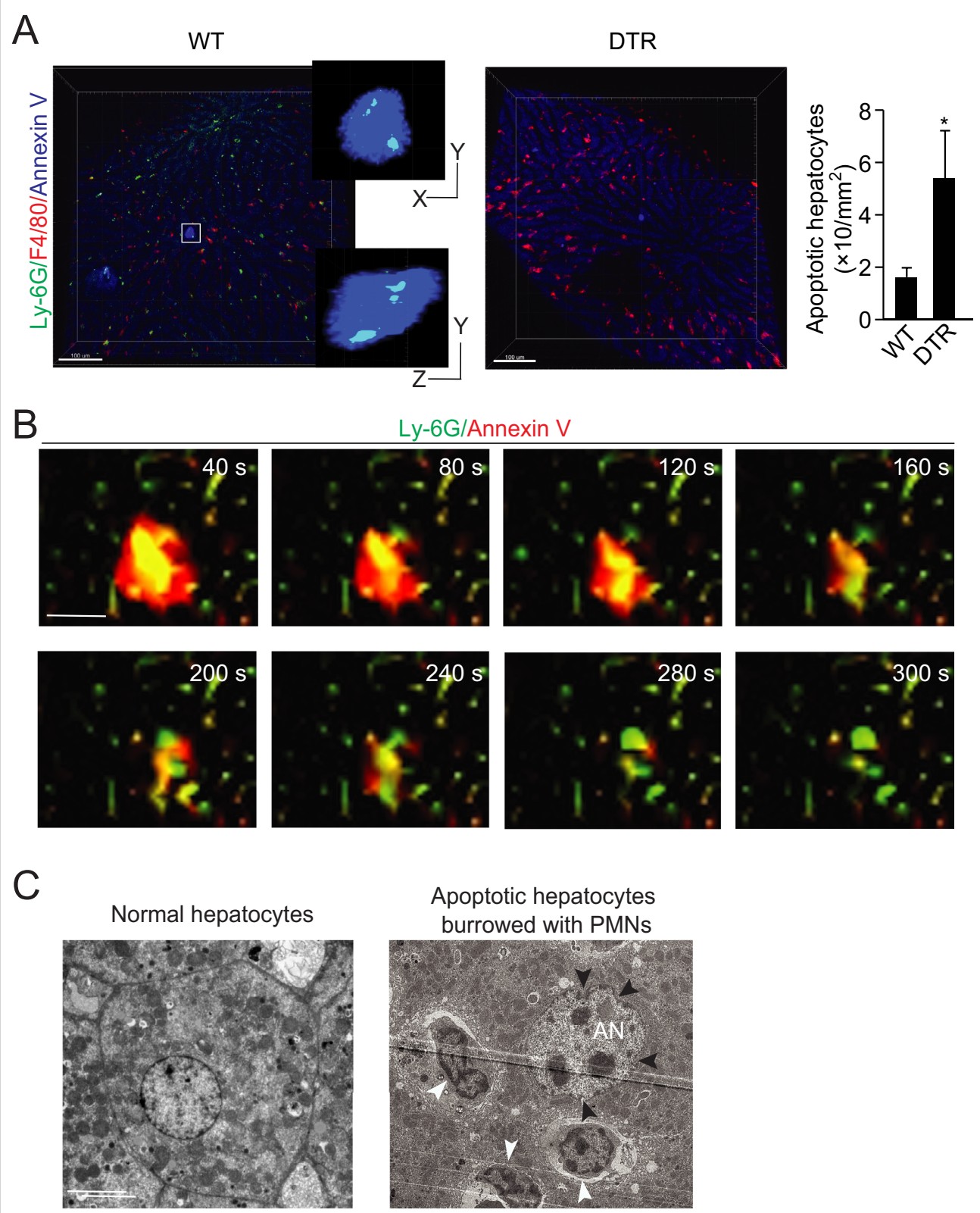

**Figure 2.** Neutrophil burrowing into apoptotic hepatocytes. (**A**) Intravital microscopy images of liver tissues from wild-type (WT) and MRP8cre/DTR mice. Neutrophils are labeled with an i.v. injection of anti-Ly6G antibody (green), KCs are labeled with anti-F4/80 antibody (red), and apoptotic cells are labeled with Annexin V (blue). Neutrophils inside and KCs associated with apoptotic hepatocytes are detected and analyzed with IMARIS software as described in Methods. The distances from neutrophils to the apoptotic hepatocyte border are recorded in *Table 6*. A total of 24 apoptotic cells were

*Figure 2 continued on next page*

*Figure 2 continued*

observed in the WT liver with an average of two burrowed neutrophils. Scale bar, 100 μm. (**B**) Intravital image sequences of neutrophils phagocytosing apoptotic hepatocytes in mouse livers at indicated time points. Neutrophils are labeled with an i.v. injection of anti-Ly6G antibody (green), and apoptotic cells are labeled with Annexin V (red). A total of 13 apoptotic cells with burrowed neutrophils were observed in 12 WT mouse livers. Scale bar, 20 μm. (**C**) Electron microscopy images of apoptotic mouse hepatocytes occupied by neutrophils. The apoptotic hepatic nucleus (AN) is evident by distorted nuclear membrane (pointed by black arrowheads). The neutrophils are indicated by white arrowheads with a characteristic multilobed nucleus. 29 apoptotic cells with burrowed neutrophils were observed. Scale bar, 5 μm. Data are representative of three independent experiments.

## The signals that attract neutrophils to apoptotic hepatocytes

To identify the signals that attract neutrophils to apoptotic hepatocytes, we screened for cytokine and chemokine secreted by NCTC cells before and after apoptosis. Apoptotic NCTC cells showed markedly increased secretion of the cytokines IL-1β, IL-6, IL-8, and IL-12 compared with nonapoptotic controls (*Figure 4A–D*). In contrast, GM-CSF, IFN-γ, TNF-α, IL-2, and IL-10 were not significantly

**Table 6.** Analysis of neutrophils burrowing inside apoptotic hepatocytes from mouse liver samples. Intravital images of neutrophils inside apoptotic hepatocytes (as shown in *Figure 2A*) are remodeled and analyzed with IMARIS software: Annexin V blue staining is set as the source channel to detect hepatic cell boundary, and Ly-6G green staining is set as the source channel to detect neutrophils inside blue cells. Only neutrophils inside hepatic cells are detected and calculated. The cell positions of both neutrophils and hepatocytes, and the distances from neutrophils to the hepatocyte border are recorded below (a total of 10 apoptotic hepatocytes are analyzed).

| Apoptotic hepatocytes | Hepatocyte position X,Y,Z (μm) | PMNs inside hepatocytes | PMN position X,Y,Z (μm) | PMN distance to hepatocyte border (μm) |
|---|---|---|---|---|
| #1 | 33772.8, 25874.0, 3077.5 | #1 | 33771.5, 25882.7, 3085.1 | 42.2 |
| | | #2 | 33768.9, 25877.5, 3083.2 | 44.1 |
| | | #3 | 33780.6, 25864.5, 3071.5 | 55.8 |
| #2 | 13583.7, 17933.4, 3955.6 | #1 | 13586.7, 17935.1, 3956.1 | 6.16 |
| | | #2 | 13581.7, 17932.3, 3955.5 | 6.76 |
| #3 | 13550.8, 17833.9, 3950.7 | #1 | 13551.5, 17833.9, 3950.7 | 11.6 |
| #4 | 13493.1, 18060.3, 3947.0 | #1 | 13491.3, 18060.9, 3946.8 | 15.4 |
| #5 | 13539.1, 17922.4, 3946.5 | #1 | 13539.3, 17922.8, 3946.3 | 16.0 |
| #6 | 13504.1, 17853.5, 3946.2 | #1 | 13504.1, 17853.3, 3946.1 | 16.2 |
| #7 | 13440.5, 18190.0, 3941.9 | #1 | 13440.9, 18189.2, 3941.4 | 20.9 |
| #8 | 13368.4, 18069.1, 3938.1 | #1 | 13369.0, 18068.6, 3938.0 | 24.3 |
| #9 | 13034.7, 18056.6, 3909.4 | #1 | 13034.6, 18054.8, 3909.7 | 24.8 |
| | | #2 | 13034.9, 18060.0, 3908.2 | 24.9 |
| #10 | 13019.5, 18118.2, 3902.9 | #1 | 13019.5, 18118.7, 3902.2 | 9.7 |

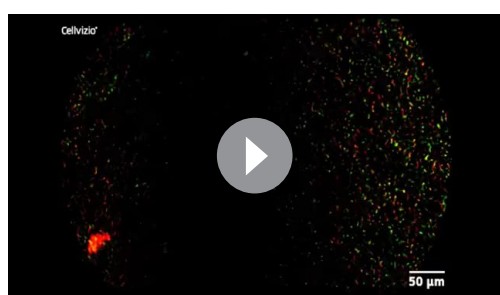

**Video 1.** Mouse neutrophils burrowed and digested apoptotic hepatocytes in the mouse liver. Neutrophils were labeled with an i.v. injection of anti-Ly6G antibody (green) and apoptotic cells were labeled with Annexin V (red). Time-lapse images were acquired by the Cellvizio system. The cytoplasm of apoptotic hepatocyte with neutrophil burrowing quickly dwindled and finally disappeared. Two out of 13 apoptotic hepatocytes are shown in Video 1 and *Video 2*.

https://elifesciences.org/articles/86591/figures#video1

changed (*Figure 4—figure supplement 1A*). To address the role of the upregulated cytokines, we knocked down the receptors of IL-1β, IL-6, IL-8, or IL-12 in HL60 cells and then examined phagocytosis ability. Knockdown of the IL-1β receptor abolished the phagocytosis of apoptotic NCTC cells (*Figure 4E*), whereas knockdown of the IL-8 receptor showed a 50% reduction in phagocytosis (*Figure 4E*). Thus, neutrophil chemoattractants, IL-1β and IL-8, secreted by apoptotic NCTC cells are key signals for attracting neutrophils and inducing subsequent phagocytosis. Consistent with above observations, we found endogenous IL-1β associated with apoptotic hepatocytes but not with normal hepatocytes in human liver samples (*Figure 4F*).

Next, we examined the cell surface molecules in hepatocytes and found that apoptotic NCTC cells had markedly increased P-selectin, E-selection, L-selectin, and PECAM as compared with nonapoptotic controls (*Figure 4G*). Inhibition of selectins (with a pan-selectin antagonist, bimosiamose) but not PECAM (with a function blocking antibody) abrogated the neutrophil-mediated clearance of apoptotic hepatocytes (*Figure 4E*), indicating selectins play a critical role during this process.

## Neutrophil depletion impairs the clearance of apoptotic hepatocytes

We next examined whether neutrophil depletion with either an antibody or genetic methods (see Methods) influences the clearance of apoptotic hepatocytes. Antibody or genetic depletion yielded about 90% or 70% reduction in mouse peripheral blood neutrophils, respectively (*Figure 5—figure supplement 1A,B*). We analyzed liver samples from neutrophil-depleted and control mice by intravital microscopy (*Figure 2A*, genetic depletion) or immunostaining (*Figure 5*, antibody depletion). Similar to human liver samples, apoptotic hepatic cells in control WT mice were occupied and phagocytosed by neutrophils (*Figures 2A, 5A and B*). After neutrophil depletion, the apoptotic bodies were no longer associated with neutrophils (*Figures 2A, 5A and B*). However, we observed that macrophages were associated with apoptotic hepatocytes in neutrophil-depleted samples (*Figures 2A, 5A and C*), suggesting a compensatory role of macrophages in phagocytosing dead hepatocytes in the absence of neutrophils. The percentage of apoptotic hepatocytes in neutrophil-depleted samples was significantly increased compared to that in controls (0.92% vs 0.2%, p<0.001, *Figure 5D*). Hence, neutrophil depletion impaired the prompt clearance of apoptotic cells in the mouse liver. To rule out the effects of environmental microbes after neutrophil depletion, we also treated the neutrophil-depleted mice with antibiotic (20 mg/kg ampicillin, i.p. injected daily). We obtained similar results in neutrophil-depleted mice with or without antibiotic treatment (*Figure 5A–D*). Meanwhile, we observed impaired liver function in neutrophil-depleted livers (with increased aspartate aminotransferase and alanine aminotransferase activity and increased total or direct bilirubin level) compared with nontreated controls (*Figure 6A–D*, antibody depletion). Depletion of neutrophils had little effect on other tissues (e.g. kidney, *Figure 5—figure supplement 1C, D*).

## Defective neutrophil perforocytosis in AIL disease

Autoimmune diseases are often linked to defective clearance of apoptotic cells (*Nagata et al., 2010*; *Poon et al., 2014*). We first determined

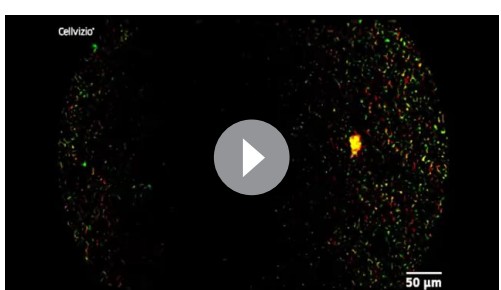

**Video 2.** Neutrophils burrow and clear apoptotic hepatocytes in vivo.
https://elifesciences.org/articles/86591/figures#video2

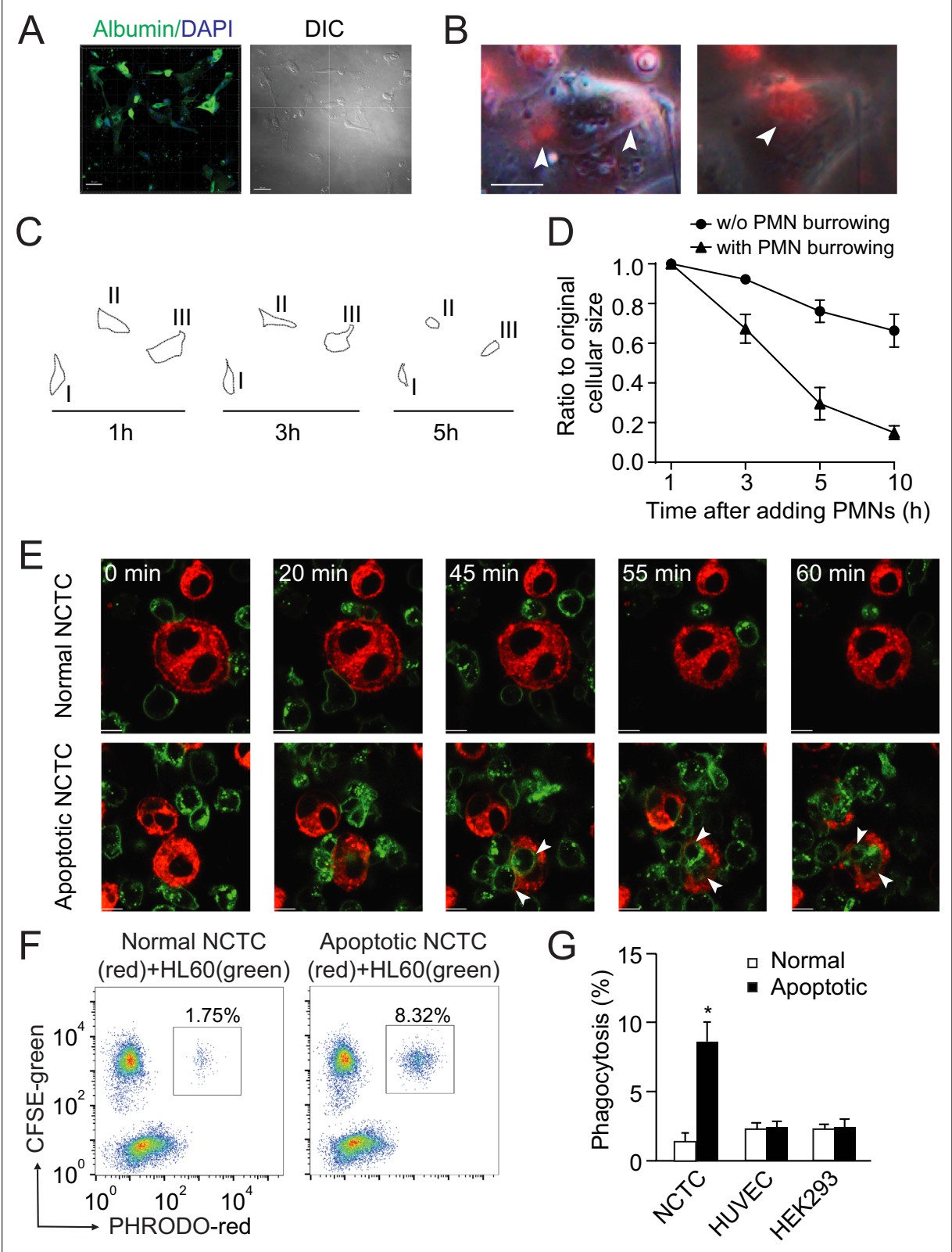

**Figure 3.** Neutrophils preferentially phagocytose apoptotic hepatocytes. (**A**) Fluorescent and phase contract images of isolated and cultured human primary hepatocytes stained with anti-albumin antibody (green). Scale bar, 50 µm. (**B**) Fluorescent and phase contract images of burrowed human neutrophils (red) inside apoptotic human primary hepatocytes from A (treated with puromycin). Scale bar, 10 µm. (**C, D**) Cell outlines (**C**) and cell size quantification (**D**) of apoptotic human primary hepatocytes occupied with burrowed neutrophils at indicated time points. (**E**) Fluorescence images of

*Figure 3 continued*

PKH67 (green)-labeled HL60 cells interacting with PKH26 (red)-labeled NCTC cells at the indicated time points. HL60 cells exhibited littler responses to nonapoptotic NCTC cells (top row). In contrast, HL60 cells polarized and penetrated apoptotic NCTC cells (bottom row). Scale bar, 10 µm. (**F**) Flow cytometry analysis of HL60 cells (labeled with CFSE, green) incubated with nonapoptotic (first column) or apoptotic hepatocytes (second column, hepatocytes were labeled with PHrodo-red dye with weak fluorescent at neutral pH but high fluorescent with a drop in pH). The population of double fluorescent HL60 cells in this subgroup (with higher PHrodo-red fluorescent indicating engulfment) was calculated, and these cells were considered phagocytosing cells. (**G**) Quantification of (**F**). HL60 cells exhibited significantly higher phagocytosis of apoptotic NCTC cells than that of nonapoptotic controls. *, p<0.05 (Student's *t*-test). Data are representative of (**A–C, E, F**) or from (**D, G**) three independent experiments, mean and sem in (D, G).

The online version of this article includes the following figure supplement(s) for figure 3:

**Figure supplement 1.** Neutrophils selectively phagocytose apoptotic liver cells.

whether defective neutrophil-mediated apoptotic clearance contributes to AIL disease. In the present study, we detected an increase in autoantibodies (i.e. against antinuclear antigen, smooth muscle actin, liver-kidney microsome, and total IgG antibodies) in neutrophil-depleted mice compared with controls (*Figure 6E–H*, antibody depletion). This increase was not affected by the treatment of antibiotic and was not observed in macrophage-depleted mice (*Figure 6E–H*, macrophages were depleted with clodronate-liposome, see Methods). Hence, neutrophil depletion not only impaired apoptotic hepatocyte clearance but also led to the generation of autoantibodies, suggesting a role of defective neutrophil-mediated removal of apoptotic bodies in AIL disease.

Next, to address whether neutrophil-mediated apoptotic clearance is impaired in AIL disease, we analyzed biopsy samples from patients diagnosed with AIL disease. In contrast to the normal human controls, a total of 22 AIL disease patient samples contained apoptotic hepatocytes that were not phagocytosed or invaded by neutrophils (*Figure 7A*). More apoptotic bodies were associated with macrophages, as observed in neutrophil-depleted mouse livers (*Figure 7A*, *Table 7*). Since the blood neutrophil count in AIL disease patients is within the normal range, we surmised potential defects in the phagocytosis ability of neutrophils from AIL disease patients. We observed markedly decreased phagocytosis of apoptotic NCTC cells by neutrophils from AIL disease patients compared to normal controls (*Figure 7B*). Normal human neutrophils burrowed into apoptotic NCTC cells and demonstrated perforocytosis, while AIL disease neutrophils exhibited little response toward apoptotic NCTC cells (*Figure 7C*). We further screened differential gene expression between normal human neutrophils and AIL neutrophils. We noted that the expression of IL-1β receptor, IL1R1 and selectin binding protein, P-selectin glycoprotein ligand 1 (PSGL-1) was markedly decreased in AIL neutrophils as compared with normal human neutrophils (*Figure 7D*), which further confirmed the critical roles of IL-1β and selectins in neutrophil-mediated apoptotic clearance. The above data prove defective neutrophil-mediated apoptotic clearance in human AIL disease samples.

## Discussion

Our finding of neutrophil burrowing into and clearing of apoptotic hepatocytes under physiological conditions reveals a fundamental mechanism for the removal of effete hepatocytes, helps to solve some mysteries surrounding the apoptotic clearance process, and raises several important questions.

### Neutrophils as the new scavengers for apoptotic clearance

Although billions of cells undergo apoptosis daily in our bodies, apoptotic cells are rarely observed in tissues under steady state due to the high efficiency of apoptotic removal by both professional and neighboring nonprofessional phagocytes

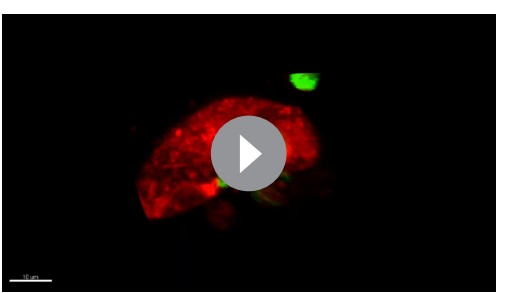

**Video 3.** Human neutrophils burrowed inside apoptotic hepatocytes. A human neutrophil (green, pointed by white arrow) was burrowing into an apoptotic hepatocyte (red). Time-lapse video was recorded for 60 min and followed by reconstructed 3D images at 20 and 60 min. At 20 min, the neutrophil partially entered the hepatic cell and did not phagocytose (neutrophil was extracted and analyzed by IMARIS software and no red dye inside the neutrophil). At 60 min, the neutrophil completely burrowed inside the hepatocyte and started to phagocytose hepatocyte (with uptake of red dye). https://elifesciences.org/articles/86591/figures#video3

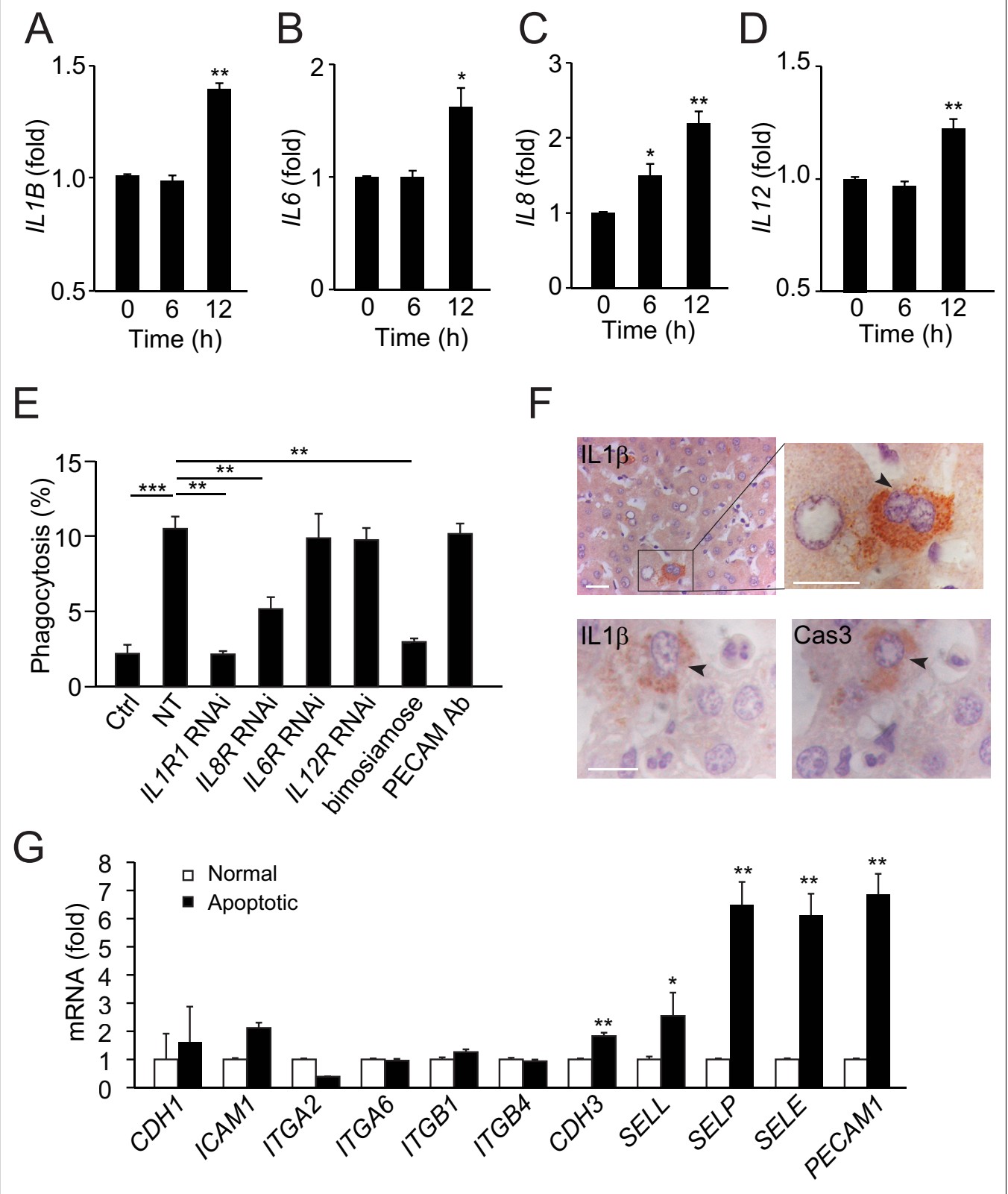

**Figure 4.** Apoptotic hepatocytes release signals that attract neutrophils for phagocytosis. (**A–D**) Cytokine secretion in apoptotic NCTC cells. IL-1β (**A**), IL-6 (**B**), IL-8 (**C**), and IL-12 (**D**) levels were significantly increased in apoptotic cells compared with nonapoptotic controls, **, $p<0.01$, ***, $p<0.001$ (Student's $t$-test). (**E**) Phagocytosis of nonapoptotic NCTC cells by HL60 cells (ctrl) and of apoptotic NCTC cells by HL60 cells treated without (NT) or with RNAi knockdown of IL-1β, IL-8, IL-6, IL-12 receptors, or with selectin antagonist (bimosiamose), PECAM blocking antibody. *, $p<0.05$ (Student's $t$-test).

*Figure 4 continued on next page*

*Figure 4 continued*

(**F**) Images of liver tissues stained with anti-IL-1β or anti-Cas-3. Top row, IL-1β is only associated with apoptotic hepatocytes with condensed chromatin (indicated by black arrowheads) but not with nonapoptotic hepatocytes. Bottom row, IL-1β and Cas-3 staining in 2 μm consecutive liver sections. Both IL-1β and Cas-3 signals are accumulated in the same apoptotic hepatocyte (indicated by black arrowhead). Scale bar, 20 μm. (**G**) Cell surface receptors in apoptotic NCTC cells compared with nonapoptotic controls. **, p<0.01 (Student's *t*-test). Data are from (**A–E, G**) or representative of three independent experiments (**F**), mean and sem in (A–E, G).

The online version of this article includes the following source data and figure supplement(s) for figure 4:

**Figure supplement 1.** Cytokines and cell surface receptors for perforocytosis.

**Figure supplement 1—source data 1.** Immunoblots of target proteins (IL-1b, IL-8, IL-6, IL-12 receptors) in nontreated (Ctrl) and RNAi-treated HL60 cells.

(*Poon et al., 2014*). In general, professional phagocytes have much higher phagocytic efficiency and capacity than nonprofessional phagocytes, but they are greatly outnumbered by other cell types in tissues (*Elliott and Ravichandran, 2016*). Therefore, whether there are unidentified scavengers and/or mechanisms for the prompt removal of dead cells in distinct organs remains unclear. Compared to macrophages or their precursors, monocytes, which make up less than 10% of the total white blood cell population, neutrophils are the most abundant white blood cells (up to 70%) in humans, and thus ideally suited for the swift clearance of apoptotic hepatic cells. As macrophages were induced to phagocytose apoptotic hepatocytes following the depletion of circulating neutrophils, our data show compensatory activation of other phagocytes under these conditions. The basis for this switch, however, is not clear.

Another important question is whether other tissues utilize a similar mechanism. As neutrophils do not phagocytose apoptotic epithelial and endothelial cells, it is possible that distinct signals are specifically released by different types of apoptotic cells. It remains largely unknown whether one or more signals are dominant in each specific tissue and how these signals are orchestrated into a complicated network to swiftly clear dying cells while avoiding tissue inflammation.

## Neutrophil influx without causing tissue damage

Although apoptotic clearance has been considered an immunologically silent process that does not lead to the influx of inflammatory cells or exposure of self-antigens (*Poon et al., 2014*; *Savill et al., 2002*), it has also been suggested that this process is not completely immunologically silent (*Green et al., 2009*). Because neutrophils are proinflammatory cells, they have long been thought to be excluded from apoptotic sites. We observed neutrophil influx into apoptotic hepatic cells (up to 22 neutrophils in one apoptotic hepatocyte), and there was little detectable tissue injury or other inflammatory cells. One reason could be that the neutrophil attraction signals released by apoptotic hepatocytes (e.g. IL-1β and IL-8) are sufficient for attracting neutrophils toward apoptotic cells and inducing subsequent perforocytosis, but do not elicit neutrophil inflammatory functions such as oxidative burst. Therefore, neutrophil recruitment into tissues does not always induce inflammation or tissue damage. In consistent with our observations, neutrophils can express both pro- and anti-inflammatory cytokines (*Gideon et al., 2019*; *Mortaz et al., 2018*). In contrast, neutrophil depletion caused defective removal of apoptotic bodies and induced autoantibody generation; thus neutrophils play a vital role in the genesis of AIL disease. Considering that billions of neutrophils patrol tissues under physiological circumstances without causing inflammation (*Nicolás-Ávila et al., 2017*), we concluded that these cells not only provide immune surveillance against infection but also contribute to internal tissue homeostasis, as we report in this study.

## Neutrophil burrowing to maintain tissue integrity

One intriguing question regarding apoptotic clearance is how the tissue maintains integrity while dead cells are being continually removed (*Poon et al., 2014*). Apoptotic cell detachment from the extracellular matrix by caspase-mediated cleavage or extrusion into the organ lumen has been proposed as the mechanism for maintaining the epithelial barrier during the clearance of dying epithelial cells (*Brancolini et al., 1997*; *Rosenblatt et al., 2001*). Our finding that burrowed neutrophils phagocytose apoptotic hepatocytes from the intracellular space provides a novel mechanism for maintaining liver integrity. The neutrophils that phagocytose cells from the inside are efficient at disposing of apoptotic

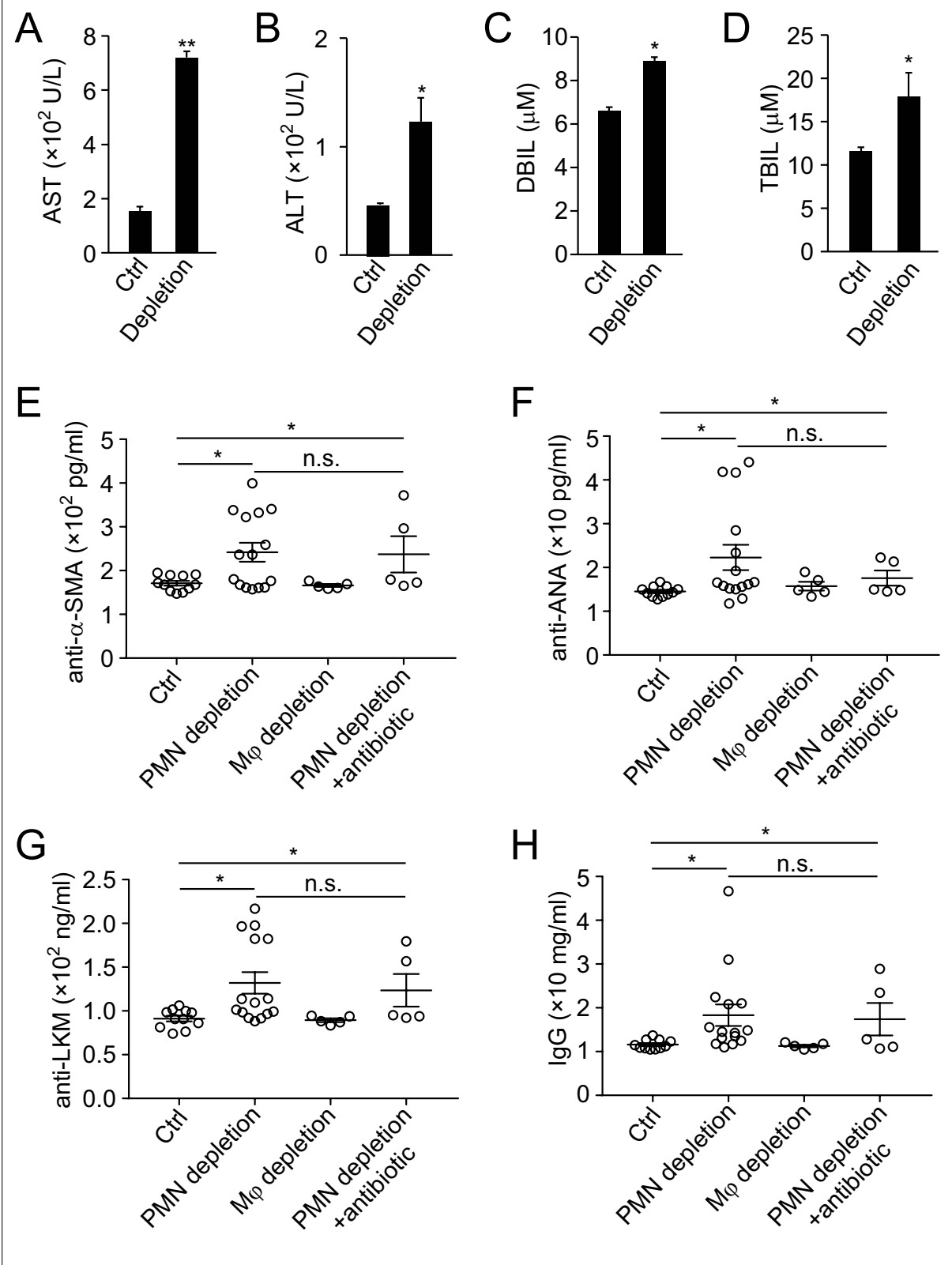

**Figure 5.** Impaired apoptotic clearance in the neutrophil-depleted liver. (**A**) Fluorescent images of liver samples from control mice (top row) or neutrophil-depleted mice (antibody depletion; without or with antibiotic, second and third rows). Neutrophils are labeled with neutrophil elastase (NE) immunostaining (purple), macrophages are labeled with F4/80 immunostaining (red), and apoptotic cells are labeled with TUNEL staining (green). Scale bar, 15 μm. (**B–D**) Cell counts of neutrophils (**B**) and macrophages (**C**) in or associated with apoptotic hepatocytes (**D**) in tissue samples as described in

*Figure 5 continued on next page*

*Figure 5 continued*

(A). *, p<0.05, **, p<0.01, compared to control (Student's *t*-test). Data are representative of (**A**) or from three independent experiments (B–D; mean and sem in B–D).

The online version of this article includes the following figure supplement(s) for figure 5:

**Figure supplement 1.** Neutrophil depletion impairs liver functions.

---

bodies without extruding the cytoplasm and may help to prevent the leakage of toxic bile acids or the release of danger signals that cause tissue damage.

Neutrophil burrowing into other types of cells has also been reported previously in processes not related to apoptotic clearance (*Overholtzer and Brugge, 2008*). For example, neutrophils can bore into endothelial cells or megakaryocytes to temporarily form so-called cell-in-cell (or emperipolesis, entosis) structures (*Overholtzer and Brugge, 2008*). The apparent difference is that both the endothelial cells and megakaryocytes entered by neutrophils are viable and nonapoptotic. The purpose of forming cell-in-cell structures with neutrophils and endothelial cells or megakaryocytes is to obtain a passage out of blood vessels or bone marrow, respectively (*Overholtzer and Brugge, 2008*). It will be interesting to examine whether neutrophils utilize similar invasion mechanisms during these processes.

The important apoptotic clearance function of neutrophils described in this study adds to the repertoire of other known functions of neutrophils (*Amulic et al., 2012*; *Kolaczkowska and Kubes, 2013*; *Wang et al., 2017*). Since the failure to clear dead cells is linked to inflammatory and autoimmune diseases, the critical role of neutrophil-mediated apoptotic clearance may have implications for the pathogenesis and treatment of these diseases.

## Methods

### Cell culture, transfection, and isolation of human and mouse neutrophils

NCTC (authenticated by STR profiling https://www.cellosaurus.org/CVCL_3066), U937 (authenticated by STR profiling https://www.cellosaurus.org/CVCL_0007) cells were cultured in DMEM supplemented with 10% fetal bovine serum and HL60 (authenticated by STR profiling https://www.cellosaurus.org/CVCL_0002) cells were cultured in RPMI 1640 with 10% fetal bovine serum. All cell lines have been tested with negative *Mycoplasma*. For experiments, HL60 cells were differentiated by adding 1.3% DMSO into the medium for 7 days (*Xu et al., 2003*). To establish the stable RNAi cell lines, we transfected shRNA into HEK293T cells. After virus packaging in HEK293T cells, HL60 cells were infected, screened by puromycin, and sorted by flow cytometry (*Liu et al., 2012*).

For mouse polymorphonuclear neutrophil isolation from bone marrow (*Wang et al., 2016*), mice were euthanized, and the femurs and tibias were removed and flushed by a 27-gauge needle with a 10 mL syringe filled with calcium- and magnesium-free Hank's balanced salt solution (HBSS) plus 0.1% BSA. Cells were then centrifuged and resuspended in HBSS. After being filtered through a 40 μm strainer, cells in 3 mL of HBSS were loaded onto a pre-prepared gradient solution (3 mL of NycoPrep on top and 3 mL of 72% Percoll on bottom). The samples were centrifuged at 2400 rpm at room temperature for 20 min with the brake off. The middle layer was collected and washed once in HBSS. Sterilized distilled water (9 mL) was added for 22 s and thereafter 1 mL of 10× PBS was added to remove red blood cells. Finally, the cells were collected and resuspended in HBSS or medium (neutrophil purity >90%). All animal experiments were performed under the protocol approved by The Institutional Animal Care and Use Committee of the State Key Laboratory of Biotherapy, West China Hospital, Sichuan University (2019222).

For the isolation of human neutrophils (*Liu et al., 2015*), blood was collected from healthy human donors or those with AIL disease. Erythrocytes were removed using dextran sedimentation (4.5% dextran) followed by hypotonic lysis using distilled water. Neutrophils were isolated from the resulting cell suspension using discontinuous Percoll gradient centrifugation. This procedure yielded >95% neutrophil purity and >95% viability as assessed by flow cytometry.

For the isolation of human primary hepatocytes, liver tissues from a fresh liver hemangioma surgery were suspended in DPBS and washed for 5–10 times. Surrounding connective tissues and adipose

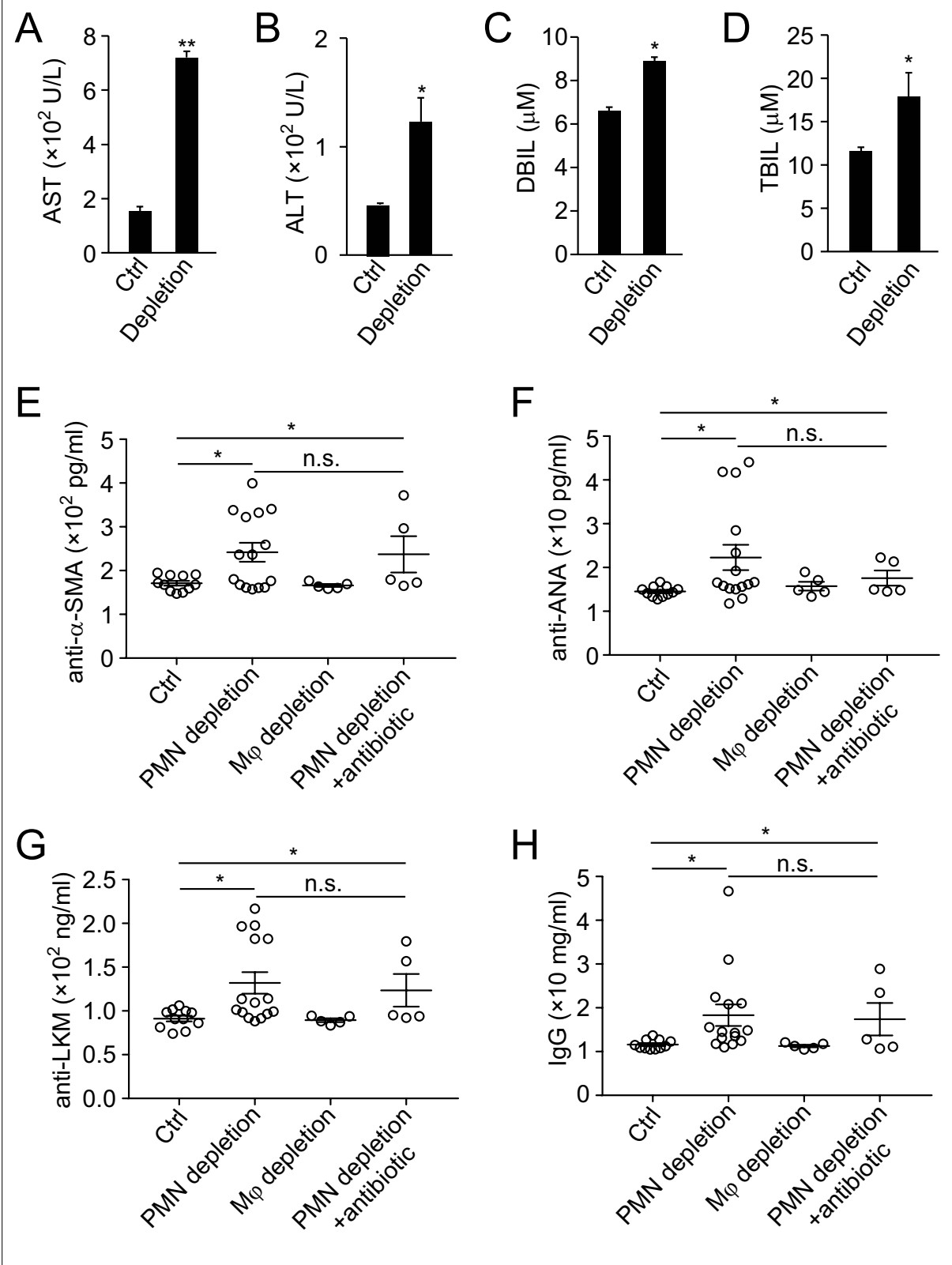

**Figure 6.** Analysis of liver function and autoantibody generation in neutrophil-depleted mice. (**A–D**) Liver function analysis of aspartate aminotransferase (AST, **A**), alanine aminotransferase (ALT, **B**), direct bilirubin (DBIL, **C**), and total bilirubin (TBIL, **D**) in ctrl or neutrophil-depleted mice (antibody depletion). *, p<0.05, **, p<0.01 (Student's *t*-test). (**E–H**) Expression of autoantibodies against smooth muscle actin (anti-α-SMA, **E**), antinuclear antigen (anti-ANA, **G**), liver-kidney microsome (anti-LKM, **G**), and total IgG (**H**) in serum of control, neutrophil-depleted (with antibody), macrophage-depleted, or neutrophil-depleted plus antibiotic-treated mice. Data are from three independent experiments (A–H; mean and sem in A–H).

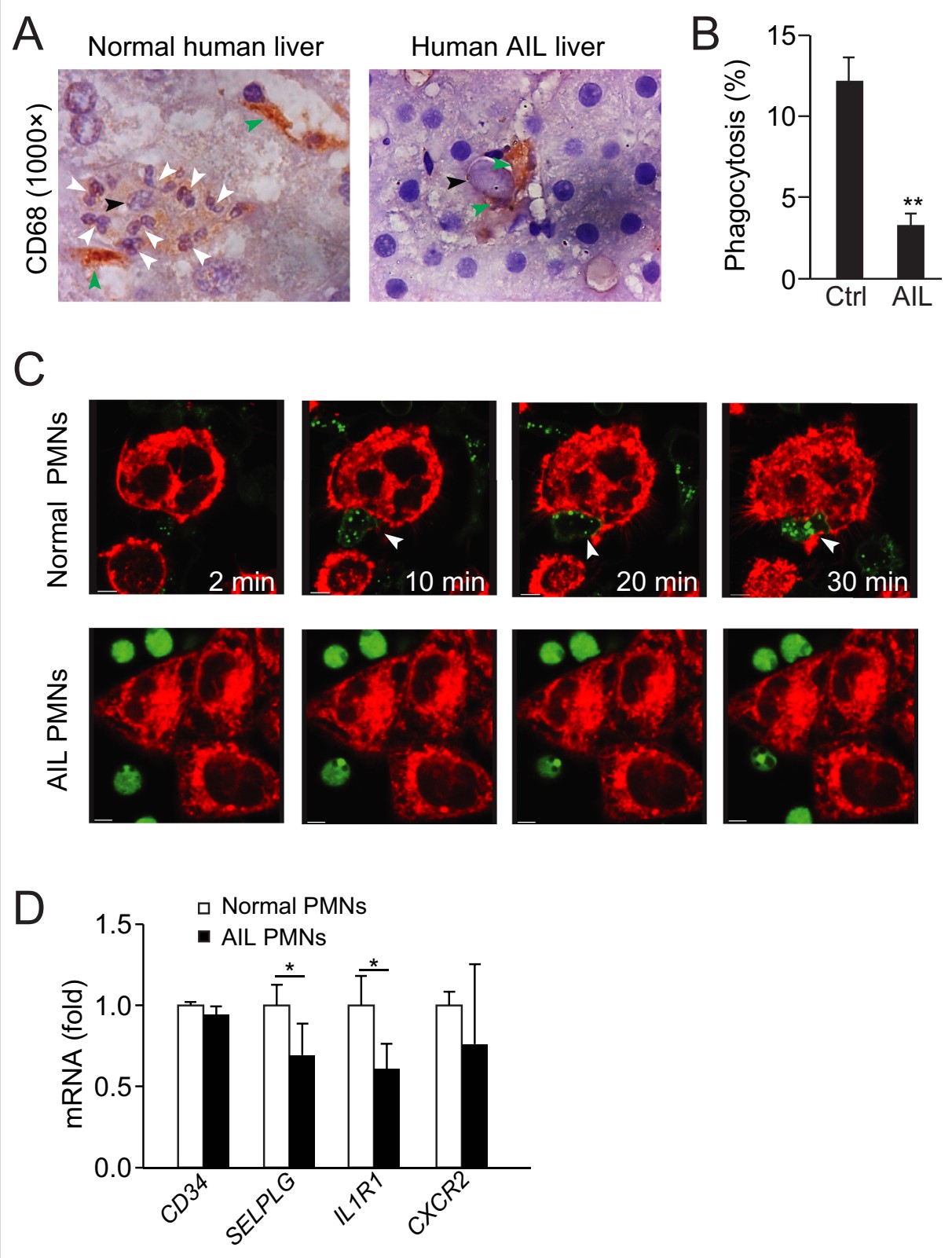

**Figure 7.** Defective neutrophil perforocytosis in human autoimmune liver (AIL) disease. (**A**) Images of CD68 staining from normal or AIL disease human liver samples. Green arrowheads indicate macrophages, black arrowheads indicate apoptotic bodies, and white arrowheads point to neutrophils. (**B, C**) Quantification (**B**) and images (**C**) of in vitro phagocytosis of apoptotic NCTC cells by neutrophils isolated from normal or AIL disease human patients. White arrowheads point to burrowed neutrophils. Scale bar, 10 μm. *, p<0.05 (Student's t-test). (**D**) Analysis of CD34, SELPLG, IL1R1, and CXCR2

*Figure 7 continued on next page*

*Figure 7 continued*

expression in neutrophils from normal human and AIL patients by microarray. Data are representative of (**A, C**) or from three independent experiments (B, D; mean and sem in B, **D**).

The online version of this article includes the following source data for figure 7:

**Source data 1.** Analysis of gene expression in neutrophils from normal human and autoimmune liver (AIL) patients by microarray.

tissues were removed with a surgery knife and remaining liver tissues were further sliced into small fragments (around 1 mm$^2$), then washed with DPBS for three times, and centrifuged at 800 rpm for 3 min. Tissue samples were resuspended in five times volume of collagenase IV medium and incubated at 37°C for 30 min. After collagenase digestion, samples were centrifuged at 1000 rpm for 5 min, and resuspended in RPMI 1640 with antibiotics to precipitate cells, then centrifuged at 1500 rpm for 5 min. Isolated cells were cultured in RPMI 1640 medium with insulin (1: 1000), glucagon (1:1000), glucocorticoid (1:1000), HGF (10 ng/mL), EGF (10 ng/mL). Studies using human primary cells and samples were approved by the Institutional Review Board of Guangzhou Regenerative Medicine and Health Guangdong Laboratory (GDL-IRB2021-011) and Fudan University (B2020-177R).

## Intravital microscopy, immunostaining of liver sections, 3D image reconstruction, and electron microscopy

For intravital microscopy (*Marques et al., 2015*), freshly prepared anti-Ly6G 1A8 FITC, anti-F4/80 PE, and Annexin V pacific blue were injected intravenously into mice 1 hr before imaging. After anesthesia, mice were attached to a surgical stage and subjected to midline laparotomy, and all vessels in the skin were cauterized. The falciform ligament between the liver and diaphragm was subsequently cut by holding the xiphoid with a knot made of suture thread. Mice were then moved to the imaging stage, and the right lobe was exposed. The ventral side of the mouse was placed on a Plexiglas stage attached to the microscope stage of a single photon microscope (NiKON A1RMP) or Cellvizio System for the imaging of apoptotic cells (labeled with Annexin V), Kupffer cells (anti-F4/80), and neutrophils (anti-Ly-6G 1A8). Microscope acquisition settings were as follows: 405 nm laser power, 15.4%, PMT high voltage (HV), 115; 488 nm laser power, 27.9%, PMT HV, 76; 543 nm laser power, 41.4%, PMT HV, 108; pinhole size, 40 μm; 20× objective.

Neutrophils burrowing inside apoptotic hepatocytes were analyzed by IMARIS software (adopted from the analysis of vesicles in cells). Annexin V blue was set as the source channel to detect apoptotic hepatocyte boundary, and anti-Ly6G green was set as the source channel to detect neutrophils inside blue cells. Only neutrophils inside hepatic cells can be detected. The cells objects were calculated and displayed. The cell positions of both neutrophils and hepatocytes, and the distances from neutrophils to the hepatocyte border were recorded.

For immunostaining of liver sections, fresh mouse liver samples were fixed in freshly prepared 4% PFA and cryoprotected in 30% sucrose/PBS solution. Thick cryosections (45 μm) were cut and washed three times with PBS. Tissue sections were incubated in blocking buffer (10% donkey serum and 0.2% Triton X-100 in PBS) for 1 hr at room temperature and then in a primary antibody solution (5% donkey serum and 0.2% Triton X-100 in PBS; rat anti-Ly-6G antibody, 1:500; goat anti-E-cadherin antibody, 1:200) overnight at 4°C. Next, the sections were washed three times in PBS, incubated for 1 hr in the appropriate secondary antibody solution containing DAPI for 2 hr at 4°C, and then washed three times in PBS. The immunostained sections were mounted onto slides with an aqueous mounting

**Table 7.** Apoptotic hepatocytes burrowed with polymorphonuclear neutrophils (PMNs) or monocytes in normal human or autoimmune liver (AIL) liver samples.
45 μm thick tissue sections were used to stain and count apoptotic hepatocytes.

| | Total liver sample numbers | Apoptotic hepatocytes | Apoptotic hepatocytes burrowed by PMNs | Apoptotic hepatocytes burrowed by monocytes |
|---|---|---|---|---|
| Normal liver samples | 32 | 227 | 227 | 0 |
| AIL liver samples | 22 | 110 | 8 | 40 |

medium. All tissue sections were scanned by a Zeiss 710 confocal microscope. The Z-stack scanned images were edited by Zen 2009 software.

To observe the liver ultrastructure with TEM, mouse livers were removed, fixed (2.5% glutaraldehyde in 0.1 M cacodylate buffer, pH 7.4), and cut into multiple 1 mm × 1 mm strips in fluid; several strips were simultaneously cut into 1 mm × 1 mm × 1 mm blocks for TEM. Blocks were transferred to wash buffer (0.1 M cacodylate buffer, pH 7.4) to remove glutaraldehyde and postfixed for 1 hr in 1% osmium tetroxide. The specimens were dehydrated in a graded series of ethanol (75%, 85%, 95%, and 100%) and embedded in Epon812. Ultrathin sections were obtained and then observed with a HITACHI H600 TEM. Apoptotic liver cells were determined based on changes in nuclear morphology (chromatin condensation and fragmentation). Neutrophils were determined by polymorphonuclear morphology.

## Neutrophil depletion and macrophage depletion in mice

Two methods were used to deplete neutrophils in mice. For antibody-based depletion, WT mice were injected i.p. with anti-mouse Ly6G clone 1A8 (Bio X Cell) supernatant containing 0.5 mg protein every 48 hr for 1 month. Control animals were injected with isotype-matched normal control antibody. For genetic neutrophil ablation, diphtheria toxin receptor (DTR)-expressing Rosa26iDTR mice (*Buch et al., 2005*; *Saito et al., 2001*) (The Jackson Laboratory) were bred onto the *Mrp8*-Cre genetic background (The Jackson Laboratory). Cre expression in MRP-expressing cells, mainly granulocytes, will activate DTR expression in only MRP-expressing cells. At the age of 3 months, the Rosa26iDTR/Mrp8-Cre mice and WT control littermates (Rosa26iDTR) were treated with diphtheria toxin (DT, 20 ng/g BW, i.p., daily for 3 days) to induce Mrp8$^+$ cell depletion.

Clodronate-liposome (YEASEN, 40337ES05) was used to deplete mouse macrophage. WT mice were injected i.v. with 100 µL of clodronate-liposome per 10 g of body weight. 24 hr later, mice were injected i.v. with 100 µL of clodronate-liposome per 10 g of body weight again. The operation was repeated every 72 hr after the second injection for 1 month. Neutrophil and macrophage depletion were confirmed by neutrophil and macrophage count in all experimental animals.

## Apoptosis assay

Apoptosis of NCTC cells was induced by puromycin (*Chow et al., 1995*). A total of 1×10$^6$ NCTC cells were plated in a 35 mm glass-bottom dish (MatTek) for 10 hr, and then treated with puromycin (2.5 µg/mL) for 12 hr. Apoptosis was assayed with an Alexa Fluor 488 Annexin V/Dead Cell Apoptosis Kit (Thermo Fisher) and analyzed by flow cytometry to measure the fluorescence emission at 530 nm and 575 nm (or equivalent) using 488 nm excitation.

## Fluorescent labeling of cells

The fluorescent membrane stain PKH-26 (red) (Sigma-Aldrich) was used to label nonapoptotic and apoptotic NCTC and HEK293 cells and HUVECs. PKH-67 (green) was used to label HL60 cells and human neutrophils. Cells (2×10$^6$) were washed in serum-free DMEM or RPMI 1640 medium, and the cell pellet was resuspended in PKH-26 or PKH-67 staining solution and incubated for 5 min at room temperature. After stopping the labeling reaction by adding an equal volume of serum, the cells were washed three times with complete medium.

## In vitro phagocytosis assay

Phagocytosis was quantified by flow cytometry using a pH sensitive dye, PHrodo-red (weak fluorescent at neutral pH but high fluorescent with a drop in pH) to measure engulfment. 1×10$^6$ human neutrophils or differentiated HL60 cells were labeled with CFSE green and then incubated with apoptotic or nonapoptotic NCTC cells (labeled with PHrodo-red, 10 µL PHrodo-red dye mixed with 100 µL PowerLoad concentrate, and diluted into 10 mL of HBSS, PH 7.4 for 1 hr). The cells were then washed with HBSS and analyzed by flow cytometry. The forward scatter reflects cell size, whereas the sideward scatter reveals the degree of granularity of the cell. Apoptotic NCTC cells with burrowed neutrophils

increased in cell size and granularity, causing increased diffraction of the laser beam, revealed by spreading of the dots to the upper right corner. The population of double fluorescent HL60 cells in this subgroup was calculated, and these cells were considered phagocytosing cells.

## Cytokine screening and microarray

Cytokines secreted from nonapoptotic and apoptotic NCTC cells were screened with the Human Inflammatory Cytokines Multi-Analyte ELISArray Kit (QIAGEN). Nonapoptotic and apoptotic NCTC cell supernatants were prepared in replicate serial dilutions of the Antigen Standard. Cytokine secretion was further determined with the ELISA kit.

For microarray, total RNA from normal human or AIL patient neutrophils was isolated using RNeasy Total RNA Isolation Kit (QIAGEN, GmbH, Germany)/TRIzol reagent (Life Technologies, Carlsbad, CA, USA) according to the manufacturer's instructions, and purified by using RNeasy Mini Kit (QIAGEN, GmbH, Germany). Total RNA was checked for a RIN number to inspect RNA integration by an Agilent Bioanalyzer 2100 (Agilent Technologies, Santa Clara, CA, USA). RNA samples of each group were then used to generate biotinylated cRNA targets for the Agilent SurePrint Gene Expression Microarray. The biotinylated cRNA targets were then hybridized with the slides. After hybridization, slides were scanned on the Agilent Microarray Scanner (Agilent Technologies, Santa Clara, CA, USA). Data were extracted with Feature Extraction software 10.7 (Agilent Technologies, Santa Clara, CA, USA). Raw data were normalized by Quantile algorithm, R package 'limma'. Heatmap plots were done by a R package 'pheatmap' of the target genes. GO/pathway enrichment analysis was done using Fisher's exact test by R package 'clusterProfiler' of the target genes. GO categories/pathway with Fisher's exact test p-values <0.05 were selected.

## Statistical analysis

All experiments were performed at least three times. Representative data are shown in the paper. An independent experiment's two-tailed Student's *t*-test was used as the statistical assay for comparisons. Significant differences between samples were indicated by p < 0.05.

## Acknowledgements

We thank Yu Fu and Ruixue Li (both from Guangzhou Regenerative Medicine and Health Guangdong Laboratory Imaging Center) for helpful imaging service. This work was supported in part by NSFC grant 32170750 and SSTP grant 19YYJC2572.

## Additional information

### Funding

| Funder | Grant reference number | Author |
|---|---|---|
| National Natural Science Foundation of China | 32170750 | Jingsong Xu |
| SSTP | 19YYJC2572 | Jingsong Xu |

The funders had no role in study design, data collection and interpretation, or the decision to submit the work for publication.

### Author contributions

Luyang Cao, Juan Zhao, Conceptualization, Data curation, Software, Formal analysis, Validation, Investigation, Visualization, Methodology, Writing – original draft, Writing – review and editing; Lixiang Ma, Xiangyu Wang, Shengxian Peng, Li Yang, Li Li, Xiaobo Hu, Yuan Ji, Xianming Zhang, Investigation, Methodology; Xinzou Fang, Wei Li, Yawen Qi, Yingkui Tang, Jieya Liu, Investigation, Visualization, Methodology; Liangxue Zhou, Yi Zhao, Yong Zhao, Yuquan Wei, Resources, Methodology; Yingyong Hou, Methodology; You-yang Zhao, Resources, Investigation, Methodology; Asrar B Malik, Resources, Methodology, Writing – original draft, Writing – review and editing; Hexige Saiyin, Conceptualization, Data curation, Formal analysis, Supervision, Validation, Investigation, Visualization,

Methodology, Writing – original draft, Project administration, Writing – review and editing; Jingsong Xu, Conceptualization, Resources, Data curation, Formal analysis, Supervision, Funding acquisition, Validation, Investigation, Visualization, Methodology, Writing – original draft, Project administration, Writing – review and editing

## Author ORCIDs
Xiangyu Wang (ID) http://orcid.org/0000-0001-8605-0727
Li Li (ID) http://orcid.org/0000-0002-2374-5203
Jingsong Xu (ID) https://orcid.org/0000-0001-8261-9509

## Ethics
Studies using human primary cells and samples were approved by the Institutional Review Board of Guangzhou Regenerative Medicine and Health Guangdong Laboratory (GDL-IRB2021-011) and Fudan University (B2020-177R).
All animal experiments were performed under the protocol approved by The Institutional Animal Care and Use Committee of the State Key Laboratory of Biotherapy, West China Hospital, Sichuan University (2019222).

Reviewer #1 (Public Review): https://doi.org/10.7554/eLife.86591.3.sa1
Reviewer #2 (Public Review): https://doi.org/10.7554/eLife.86591.3.sa2
Author Response: https://doi.org/10.7554/eLife.86591.3.sa3

## Additional files

### Supplementary files
• MDAR checklist

### Data availability
All data generated or analyzed during this study are included in the manuscript and supporting file; Source data files have been provided for Figure 4—figure supplement 1 and Figure 7.

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

# Appendix 1

## Appendix 1—key resources table

| Reagent type (species) or resource | Designation | Source or reference | Identifiers | Additional information |
|---|---|---|---|---|
| Strain, strain background C57BL/6 (mouse M) | C57BL/6-Gt(ROSA)26Sor$^{tm1(HBEGF)Awai}$/J | The Jackson Laboratory | 007900 | |
| Strain, strain background B6.cg (mouse M) | B6.Cg-Tg(S100A8-cre,-EGFP)1Ilw/J, Mrp8creTg | The Jackson Laboratory | 021614 | |
| Strain, strain background B6.129 (mouse M) | B6.129S7-Il1r1$^{tm1Imx}$/J | The Jackson Laboratory | 003245 | |
| Cell line (human) | HL-60 | ATCC | CCL-240 | https://www.cellosaurus.org/CVCL_0002 |
| Cell line (human) | NCTC | ZQXZBIO | ZQ0631 | https://www.cellosaurus.org/CVCL_3066 |
| Cell line (human) | U937 | ATCC | CRL-1593 | https://www.cellosaurus.org/CVCL_0007 |
| Biological sample (human) | Freshly surged human liver from angioma | Huashan Hospital | Dr. Wang Xiangyu | |
| Biological sample (human) | Human hepatocellular patient tissue slides | Zhongshan Hospital | Dr. Ji Yuan | |
| Biological sample (human) | Human paraffin tissue slides from hemangioma patients | Zhongshan Hospital | Dr.Ji Yuan | |
| Biological sample (human) | Freshly surged liver cancer samples | Zhongshan Hospital | Dr. Ji Yuan | |
| Biological sample (human) | Human Blood sample | Longhua Hospital | Dr. Xiaobo Hu | |
| Antibody | Anti-Neutrophil Elastase (Mouse monoclonal) | Abcam | ab131260 | 1:200 |
| Antibody | Anti-E Cadherin antibody (Mouse monoclonal) | Abcam | ab40772 | 1:1000 |
| Antibody | Anti-E Cadherin antibody (Mouse monoclonal) | Abcam | ab76319 | 1:100 |
| Antibody | Cleaved Caspase-3 (Asp175) Antibody (Mouse monoclonal) | Cell Signaling | #9661 | 1:500 |
| Antibody | Anti-Caspase-3 antibody (Mouse monoclonal) | Abcam | ab13847 | 1:200 |
| Antibody | Anti-Myeloperoxidase antibody (Mouse monoclonal) | Abcam | ab9535 | 1:1000 |
| Antibody | Anti-CD68 antibody (Mouse monoclonal) | Abcam | ab955 | 1:1000 |
| Antibody | Anti-CD11b antibody (Mouse monoclonal) | Abcam | ab133357 | 1:2000 |
| Antibody | CD45RA antibody (Rabbit polyclonal) | BD | 337167 | 1:2000 |
| Antibody | Anti-Ly6G antibody (Rabbit polyclonal) | BD | 551460 | 1:1000 |
| Antibody | Anti-F4/80 antibody (Rabbit polyclonal) | BD | 565410 | 1:1000 |
| Antibody | IL-1RI Antibody (Rabbit polyclonal) | SCBT | sc-393998 | 1:1000 |
| Antibody | IL-6Rα Antibody (H-7) (Rabbit polyclonal) | SCBT | sc-373708 | 1:1000 |
| Antibody | IL-8RA Antibody (Rabbit polyclonal) | SCBT | sc-7303 | 1:1000 |
| Antibody | IL-12Rβ1 Antibody (Rabbit polyclonal) | SCBT | sc-365395 | 1:1000 |
| Antibody | PECAM Antibody (Rabbit polyclonal) | Affinity | AF6191 | 1:1000 |
| Antibody | InVivoPlus anti-mouse Ly6G (Rabbit polyclonal) | Bio X Cell | BP0075-1 | 1:1000 |

*Appendix 1 Continued on next page*

*Appendix 1 Continued*

| Reagent type (species) or resource | Designation | Source or reference | Identifiers | Additional information |
|---|---|---|---|---|
| Recombinant DNA reagent | IL-1RI shRNA plasmid | SCBT | sc-35651-SH | |
| Recombinant DNA reagent | IL-6Rα shRNA plasmid | SCBT | sc-35663-SH | |
| Recombinant DNA reagent | IL-8RA shRNA plasmid | SCBT | sc-40026-SH | |
| Recombinant DNA reagent | IL-12Rβ1 shRNA Plasmid | SCBT | sc-35649-SH | |
| Commercial assay or kit | In Situ Cell Death Detection Kit | Roche | 11684795910 | |
| Commercial assay or kit | PKH26 Red Fluorescent Cell Linker Kit | Sigma | PKH26GL | |
| Commercial assay or kit | PKH67 Green Fluorescent Cell Linker Kit | Sigma | PKH67GL | |
| Commercial assay or kit | Quant-iT PicoGreen dsDNA | Invitrogen | P11495 | |
| Commercial assay or kit | Total Superoxide Dismutase Assay Kit | Beyotime | S0101S | |
| Commercial assay or kit | Reactive Oxygen Species Assay Kit | Beyotime | S0033S | |
| Commercial assay or kit | Multi-Analyte ELISArray Kits | QIAGEN | MEM-004A | |
| Commercial assay or kit | Multi-Analyte ELISArray Kits | QIAGEN | MEH-004A | |
| Commercial assay or kit | Mouse ALT ELISA Kit | mlbio | ml063179-J | |
| Commercial assay or kit | Mouse AST ELISA Kit | mlbio | ml058659 | |
| Commercial assay or kit | Mouse DBIL ELISA Kit | mlbio | ml037215 | |
| Commercial assay or kit | Mouse TB ELISA Kit | mlbio | ml037201 | |
| Commercial assay or kit | Mouse Creatinine ELISA Kit | mlbio | ml037580 | |
| Commercial assay or kit | Mouse Blood urea ELISA Kit | mlbio | ml057588 | |
| Commercial assay or kit | Mouse anti-ANA ELISA Kit | mlbio | ml002245 | |
| Commercial assay or kit | Mouse IgG ELISA Kit | mlbio | ml037601-J | |
| Commercial assay or kit | Mouse anti-LKM ELISA Kit | mlbio | ml025842 | |
| Commercial assay or kit | Mouse anti-α-SMA ELISA Kit | mlbio | ml002066 | |
| Chemical compound, drug | Annexin V | Abbkine | KTA0001 | |
| Chemical compound, drug | Puromycin | Gibco | A1113803 | |
| Chemical compound, drug | pHrodo Red AM | Invitrogen | P35372 | |
| Chemical compound, drug | Bimosiamose | MCE | TBC-1269 | |
| Chemical compound, drug | Ampicillin | Sigma | A9518 | |
| Chemical compound, drug | Clodronate-liposome | YEASEN | 40337ES05 | |
| Chemical compound, drug | α-Galactosylceramide | Abcam | ab144262 | |
| Chemical compound, drug | FasL | R&D Systems | 6128-SA | |

*Appendix 1 Continued on next page*

*Appendix 1 Continued*

| Reagent type (species) or resource | Designation | Source or reference | Identifiers | Additional information |
|---|---|---|---|---|
| Chemical compound, drug | NycoPrep | Progen | 1114550 | |
| Chemical compound, drug | Percoll | GE Healthcare | 17-0891-01 | |
| Software, algorithm | Imaris | Bitplane | https://imaris.oxinst.com/ | |
| Software, algorithm | FlowJo | BD Biosciences | https://www.flowjo.com/ | |
| Software, algorithm | GraphPad Prism | GraphPad | https://www.graphpad.com/scientific-software/prism/ | |
| Software, algorithm | ZEN Digital Imaging for Light Microscopy | ZEISS | https://www.zeiss.com | |

