## [Editor Report · eLife assessment]

This paper reports the **fundamental** discovery of a new function of neutrophil in specifically clearing apoptotic hepatocytes by penetrating the cells rather than engulfing them without causing inflammation as a part of tissue homeostasis. This **solid** study transforms the way we think about role of neutrophil in pathogenesis of autoimmune liver disease.

---

## [Referee Report · Reviewer #1 (Public Review)]

This study by Cao et al. demonstrates role of Neutrophil in clearing apoptotic hepatocytes by directly burrowing into the apoptotic hepatocytes and ingesting the effete cells from inside without causing inflammation. The authors applied intravital microscopy, Immunostaining and electron microscopy to visualize perforocytosis of neutrophil in hepatocytes. They also found that neutrophil depletion impairs the clearance of apoptotic hepatocytes causing impaired liver function and generation of autoantibodies, implying a role of defective neutrophil- mediated clearance of apoptotic cells in Autoimmune Liver disease. The experiments were well designed and conducted, the results were reasonably interpreted, and the manuscript was clearly written with logical inputs.

Further studies to explore the signals/mechanisms that determine why neutrophil specifically target apoptotic hepatocytes in liver would be of great clinical significance.

---

## [Referee Report · Reviewer #2 (Public Review)]

Neutrophils are not known to be the cells responsible for removal of apoptotic cells through efferocytosis. This report provides strong evidence that neutrophils can remove apoptotic hepatocytes in vivo and in vitro. In addition, neutrophils, which are much smaller in size than hepatocytes, can burrow into apoptotic hepatocytes.

Neutrophils are the most abundant circulating leukocytes in human. They play important roles in innate immune responses to infections and tissue injuries. Although they are dept in phagocytosis of microbes, neutrophils are not known to normally conduct efferocytosis or phagocytose host cells including apoptotic cells and play a significant role in apoptotic cell removal. In this report the authors provide evidence to suggest that neutrophils are involved in removal of apoptotic hepatocytes with certain specificity (i.e., they do not remove HEK293 or HUVEC endothelial cells). Moreover, the authors also show that neutrophils can burrow into the target cells and possibly ingest the target cells from the inside. The authors thus term this neutrophil-mediated efferocytosis process as "perforocytosis". Furthermore, evidence is provided to suggest that this neutrophil-mediated efferocytosis process keeps the number of apoptotic cells low in the livers and that defects in the processes may associate with autoimmune liver (AIL) disease phenotypes. Therefore, many of these findings are novel and the study is of important implications in our understanding of the role of neutrophils in autoimmune disease. Overall speaking, as the first report describing this novel finding, the authors have provided reasonably strong evidence for the conclusion that neutrophils burrow into apoptotic hepatocytes to perform "perforocytosis" to eliminate apoptotic hepatocytes. The importance, particularly in vivo significance, of this phenomenon needs to be further substantiated in future studies.

---

## [Author Response]

The following is the authors' response to the original reviews.

Thank you for considering our manuscript “An Unexpected Role of Neutrophils in Clearing Apoptotic Hepatocytes In Vivo". We also thank the referees for their review. We have addressed their comments in detail and added new data to buttress our conclusions.

**Reviewer #1 (Public Review):**
This study by Cao et al. demonstrates role of Neutrophil in clearing apoptotic hepatocytes by directly burrowing into the apoptotic hepatocytes and ingesting the effete cells from inside without causing inflammation. The authors applied intravital microscopy, Immunostaining and electron microscopy to visualize perforocytosis of neutrophil in hepatocytes. They also found that neutrophil depletion impairs the clearance of apoptotic hepatocytes causing impaired liver function and generation of autoantibodies, implying a role of defective neutrophil- mediated clearance of apoptotic cells in Autoimmune Liver disease. The experiments were well designed and conducted, the results were reasonably interpreted, and the manuscript was clearly written with logical inputs.

Thank you for your comments.

One weak point is that the signals/mechanisms that determine why neutrophil specifically target apoptotic hepatocytes in liver and no other organs or cells is not clearly understood.

We are still studying why neutrophils selectively phagocytose hepatocytes but not HUVEC or 293 cells. We have some intriguing preliminary data so far showing that apoptotic 293 cells have no significant increase of IL-1β production as compared with their nonapoptotic controls; both apoptotic 293 cells and HUVECs do not have increased surface selectin proteins (new Fig. S3C).

**Reviewer #2 (Public Review):**
[…] By examination of HE-stained, noncancerous liver tissue sections from patients with hepatocellular carcinoma and hepatic hemangioma, the authors observed that cells with neutrophil nuclear morphology were inside apoptotic hepatocytes. The authors also further characterized this observation by staining the sections with neutrophil and apoptosis markers. In addition, the authors observed the same phenomena in mouse livers using intravital microscopy, which also recorded the time course of the disappearance of a neutrophil-associated apoptotic cell. The author went on further characterization of neutrophil-mediated efferocytosis of cultured hepatic cells in vitro and demonstrated the process was specific for apoptotic hepatic cells, but not HEK293 or endothelial cells. The in vitro system was then used to characterize the molecular bases for neutrophil-mediated efferocytosis of apoptotic hepatic cells. The evidence was provided to suggest that IL1b and IL-8 released from and selectins upregulated in apoptotic hepatic cells were important. Importantly, the authors used two methods to deplete the neutrophils and showed that the neutrophil depletion increased apoptotic cells in livers. Finally, the authors showed that neutrophil depletion caused defects in liver function parameters. At the end, the authors presented evidence to suggest that AIL disease may be due to defective neutrophils that fail to perform "perforocytosis."

Thank you for your comments.

Point #1. Although the evidence in its totality indicates that neutrophils burrow into apoptotic hepatocytes, the significance of this "perforocytosis" phenomenon and the circumstances under which it may occur remain to be better defined. In both neutrophil depletion models, the TNUEL-positive cells were not definitively identified rather than assuming they were hepatocytes.

Anatomically, the apoptotic hepatocytes are randomly distributed in the hepatic plate from the central vein to the portal region (please refer to the image below: hematoxylin staining of liver tissues, black arrowhead indicates perforocytosis sites).

Histologically, the structure of liver/hepatic lobe are well defined, and the cell types in the livers are easy to histologically identify based on their location, morphology and the relationship to hepatic plate and sinusoid. In addition, the hepatocytes are well known for its rich cytoplasmic components, cellular connection and prominent large round nucleus. Thus, hepatocytes are very easy to identify even without using specific molecular markers such as E-cadherin or albumin. Based on these characteristics, the TUNEL positive cells that we displayed in Fig. 5A are apoptotic hepatocytes.

Point #2. In addition, there are discrepancies in the number of neutrophils and apoptotic cells in mouse liver studies; Fig. 2a WT (many neutrophils; locations unclear) vs Fig. 5A Ctr (a few neutrophils that appear in or near a vessel), and Fig. 2a DTR (a few apoptotic cells) vs Fig. 5A Depletion (many apoptotic cells).

In response, Fig. 2A demonstrates a larger area of the mouse liver (bar, 100 µm), while Fig. 5A exhibits a relatively small area of the liver sample (bars, 20 µm for Ctrl and 15 µm for DTR). Similarly, apoptotic cells in Fig. 2A DTR need to zoom in to quantify. We apologize for the confusion, and we did quantify the apoptotic cells in Fig.2A WT vs DTR (see the bar graph next to the images in Fig. 2A).

Point #3. Importantly, Fig 5a Ctrl, which is presumably a section from a mouse without any surgical treatment or without inflammation, the sole TUNNEL signal does not appear to be associated with neutrophils. Does this mean that "perforocytosis" primarily occurs in inflamed livers (Of note, human liver samples in Fig 1 are from patient with tumors. There should be inflammation in the livers of these patients).

In Fig 5A Ctrl, the TUNEL signal indicates apoptotic hepatocytes. The neutrophils (stained with anti-NE antibody, red) are associated with the apoptotic hepatocyte (Fig. 5A). We observed that perforocytosis primarily occurs in normal noninflamed livers.

Human liver samples in Fig 1 are from patient with tumors, hence it is possible that neutrophil burrowing is somehow associated with cancerous/inflammatory livers as the reviewer pointed out. This possibility was ruled out based on our method of sample preparation and experimental results themselves.

1. Both noncancerous and cancerous liver samples were sliced based on the anatomical appearance of normal and cancer tissues (differences were rather easy to identify, and these samples were prepared by highly experienced pathologists from the Liver Cancer Center of Zhongshan Hospital, Shanghai). Furthermore, the results were confirmed by determining whether the surrounding tissue contained microlesions characteristic of metastatic tumors. We only counted apoptotic hepatocytes in noncancerous regions having normal liver lobes and morphologically normal hepatocytes, plates, sinusoid and Kupffer cells. We also excluded hepatoma, chronic inflammatory regions, and necrotic regions.

2. We did not observe recruitment of neutrophils into apoptotic HCC cells, indicating that the clearance of apoptotic cancer cells was not mediated by neutrophils (unpublished observations).

3. It is hard for us to obtain normal human liver samples; however, we did study samples from patients with liver hemangioma characterized by aberrant vasculature in livers but with normal liver functions and the structure of hemangioma livers that we analyzed are nearly identical to a healthy liver in histology (these liver samples contained no cancerous regions and there was no apparent cirrhosis or inflammation). And here we obtained similar results (these are shown in Fig. 1B; a total of 40 apoptotic hepatocytes were examined).

4. Our data from normal mouse livers, isolated primary cells (hepatocytes and neutrophils) and cell lines (NCTC and HL60) all confirmed the central findings in this paper (Fig. 2, 3).

Point #4. The data on human AIL patient neutrophils raises more questions: how many AIL patients have been examined? Do these AIL neutrophils lack IL1, IL8 receptors, and/or selectin ligands? Are there increases in apoptotic hepatocytes in AIL patients?

In response, we have analyzed 16 AIL patient samples (see table below).

**Author response table 1. sa3table1:** 

No.total apototic cells	phagocytlized by PMNS	phagocytlized by Monoctye		
normal liver	32	227	227	0
AIL	16	110	8	40

We performed microarray assay to screen the differential gene expression of neutrophils from normal and liver autoimmune patients. We have identified that IL-1β receptor, IL1R1 and selectin binding protein, P- selectin glycoprotein ligand 1 (PSGL-1) were all decreased in neutrophils from the AIL patients (new Fig 7D). These findings are consistent with our observations using cells and mouse models.

Point #5. Additionally, the overall numbers of apoptotic cells even in the absence of neutrophils are rare; thus, it is questionable that such rarity of apoptotic cells can cause significant AIL phenotypes.

We quantified apoptotic liver cells in percentages instead of overall numbers (Fig. 5, we were not able to precisely calculate the overall numbers, which could be large since billions of cells undergoing apoptosis daily). Depletion of neutrophils increased the percentage of apoptotic cells about 5-6-fold in livers, and we observed the generation of autoantibodies (Fig. 6).

**Reviewer #1 (Recommendations For The Authors):**
This study by Cao et al. was well designed and conducted, the results were reasonably interpreted, and the manuscript was clearly written with logical inputs.It would further gain the significance of this study if authors could address the following questions:1. What are the mechanisms/ signals that prevents AIL Liver neutrophils from burrowing into hepatocytes?

We have identified that IL-1β receptor, IL1R1 and selectin binding protein, P-selectin glycoprotein ligand 1 (PSGL-1) were all decreased in neutrophils from the AIL patients (new Fig 7D).

1. Have authors looked if autoantigens expressed on hepatocytes, which are often found in autoimmune liver disease trigger signaling events that activate neutrophils to burrow?

Thank you for the comment, we have not examined autoantigens expressed in hepatocytes and plan to carry out this research as suggested.

1. Is perforocytosis observed in apoptotic hepatocytes induced by different agents like LPS, TNF-a , rapamycin, alcohol etc?

We did not observe perforocytosis in LPS or TNF-a treated hepatocytes. One possible reason is that LPS or TNF-a we used induced massive necrosis instead of apoptosis. Howere, we did observe neutrophil perforocytosis in FasL-induced apoptotic hepatocytes (unpublished observations).

**Reviewer #2 (Recommendations For The Authors):**
In addition to the questions raised in the "Public review" section, the authors are also recommended to address the following issues:1. Why is CD11b+ not associated with the apoptotic sites as neutrophils express CD11b

We have co-immunostained human liver samples with CD11b antibody (from Abcam: ab133357) and MPO antibody (from R&D: AF3667) and observed that tissue infiltrating neutrophils in livers have low to undetectable levels of CD11b expression (please refer the image below; white arrowheads point to neutrophils). Few CD11b+ cells in liver tissues express MPO (the CD11b+ cells are mostly macrophages, unpublished observations).

Based on these data, we conclude that CD11b is hardly expressed in neutrophils inside livers.

**Author response image 2. sa3fig2:** 

1. Can TUNEL signals in Fig. S1C be from apoptotic neutrophils?

In response, the fragmentation of nucleus is a hallmark of apoptosis hence TUNEL staining will uniformly label all fragmented parts of apoptotic nucleus. The nucleus of NE+ neutrophils are not labelled by TUNEL staining in Fig. S1C. The TUNEL+ nuclear fragments seen inside neutrophils are nuclear debris of apoptotic hepatocytes phagocytosed by neutrophils (Fig. S1C).

1. The Fig 2B experiment may be done with induced apoptosis so that neutrophil burrowing steps may be recorded from the very beginning and a better time course for the entire process can be assessed.

Thank you for the suggestions, we had tried many times with various conditions, yet still had no success to capture the very beginning of perforocytosis in vivo. We are continuing to work on this.

1. In "we found thatU937 cells exhibited much lower phagocytosis of apoptotic NCTC cells than did HL60 cells (Fig. S2B, C)," the citation should be only S2C

Thank you for pointing this out, we have corrected this in the manuscript.

1. Both neutrophil depletion models cause neutrophil death, which may complicate the interpretation of the liver function and AIL disease phenotypes. A neutropenic model such as G-CSFR−/− or Cebpe-/- mice may be used to avoid the caveat of antibody/DTR-dependent depletion models.

Thank you for this thoughtful suggestion. We have also induced AIL phenotypes in mice by using α- Galcer. α-Galcer did not cause neutrophil death but impaired neutrophil perforocytosis and futher generated AIL phenotypes in mice (unpublished observations). We plan to perform the simiarl experiments in *G-CSFR−/− or Cebpe−/−* mice as the reviewer suggested.

1. RNAi silencing experiments need additional controls for off-target effects

These RNAi silencing constructs were purchased from Santa Cruz Biotechnology and the off-target effects have been tested by the company. No significant off-target effects have been detected according to the manufacture report.